# Sampling the diurnal and annual cycles of the Earth's energy imbalance with constellations of satellite-borne radiometers

Thomas Hocking[1,2], Thorsten Mauritsen[1,2], and Linda Megner[1,2]

[1]Department of Meteorology, Stockholm University, Sweden
[2]Bolin Centre for Climate Research, Stockholm University, Sweden

**Correspondence:** Thomas Hocking (thomas.hocking@misu.su.se)

**Abstract.** The Earth's energy imbalance, i.e. the difference between incoming solar radiation and outgoing reflected and emitted radiation, is the one quantity that ultimately controls the evolution of our climate system. Despite its importance, the exact magnitude of the energy imbalance is not well known, and because it is a small net difference of about $1\ \mathrm{Wm}^{-2}$ between two large fluxes (approximately $340\ \mathrm{Wm}^{-2}$), it is challenging to measure directly. There has recently been a renewed interest in applying wide-field-of-view radiometers onboard satellites to measure the outgoing radiation, as part of methods to deduce the global annual mean energy imbalance. Here we investigate how to sample, using a limited number of satellites, in order to correctly determine the global annual mean imbalance and interannual trends. We simulate satellites in polar ($90°$ inclination), sun-synchronous ($98°$) and precessing orbits ($73°$, $82°$), as well as constellations of these types of satellite orbits. We find that no single satellite provides sufficient sampling, both globally and of the diurnal and annual cycles, to reliably determine the global annual mean. If sun-synchronous satellites are used, at least six satellites are required for an uncertainty below $1\ \mathrm{Wm}^{-2}$. One precessing satellite combined with one polar satellite results in root mean square errors of $0.08$ to $0.10\ \mathrm{Wm}^{-2}$, and a combination of two or three polar satellites results in root mean square errors of $0.10\ \mathrm{Wm}^{-2}$ or $0.04\ \mathrm{Wm}^{-2}$, respectively. In conclusion, at least two satellites that complement each other are necessary in order to ensure global coverage and achieve sampling uncertainty well below the current estimate of the energy imbalance.

# 1 Introduction

The Earth's energy imbalance (EEI) determines the current rate of accumulation of energy in the climate system, and is believed to have increased over the past decades (Kramer et al., 2021; Loeb et al., 2021; Raghuraman et al., 2021; Cheng et al., 2022; von Schuckmann et al., 2023). This net difference between incoming and outgoing radiation at the top of the atmosphere (TOA) is a fundamental property of the climate system, and can serve as a useful quantity both to indicate large-scale changes in the global energy budget over time and to better understand the climate system. The current best estimates of the TOA EEI over recent decades are 0.57 [0.43 to 0.72] $\mathrm{Wm}^{-2}$ net incoming radiation for the period 1971-2008, and the higher value 0.79 [0.52 to 1.06] $\mathrm{Wm}^{-2}$ for the period 2006-2018 (Forster et al., 2021, p. 938). In this work, we take inspiration from the Earth Climate Observatory (ECO) satellite mission proposal, and evaluate the potential of wide-field-of-view radiometers for long-term monitoring of the EEI.

Historically, analysis of the Earth's radiation balance has passed through multiple stages, from treatment of sporadic measurements, via the first organised attempts to achieve global coverage, to more numerous observations both from space and on Earth as humanity entered the satellite era. The different components of the Earth's TOA energy budget were initially investigated in the 19[th] century, but these investigations suffered from a lack of systematic global observations. The resulting values could be far from the ones we know today, notably exemplified by estimates of total solar irradiance at 1 AU (i.e. the so-called solar constant) of over 2000 $\mathrm{Wm}^{-2}$ (Shaw, 1926). Starting in the 20[th] century, improved observational data made it possible to analyse the global distribution of radiation in more detail, and also to determine the global average more accurately. Estimates of the Earth's albedo were a central part of many studies, and initial overestimates gradually approached the now established value of around 0.3 over the course of the first half of the century (Hunt et al., 1986; Goode et al., 2001). Notably, Danjon used observations of earthshine on the moon to find an albedo of 0.29 already in 1928, but later rejected this value in favour of a revised estimate to the higher value 0.39 (Danjon, 1928, 1936; Hunt et al., 1986). A more detailed overview of the history of albedo studies can be found in Stephens et al. (2015).

In the second half of the 20[th] century, the first relevant satellite missions were launched. Satellites such as the Explorer 7 launched in 1959 and the Nimbus 3 launched in 1969 used radiometers to directly measure the TOA radiation (House et al., 1986). Over the course of the following decades, satellite missions evolved from short lifetimes of months to longer lifetimes of many years. The individual satellites followed various orbits, with an overall trend over time from drifting local times to sun-synchronous or geostationary orbits (House et al., 1986). The instrument payloads also gradually changed, from initial missions that had either wide-field-of-view (WFOV) radiometers, scanning radiometers, or both, to missions that had mostly scanning radiometers.

In parallel with the continued development of direct measurements at the TOA, there have also been efforts to determine the EEI from the change in the overall heat accumulated by the Earth, notably the ocean heat content. These inventory methods rely on in-situ measurements of ocean temperatures, or e.g. satellite measurements of the ocean sea level and hence the thermal expansion of the oceans as well as contributions from melted land ice. By quantifying the change over time in this stored energy in the oceans, which take up most of the heat due to the positive EEI, it is possible to determine the corresponding average EEI

over the same period (Wong et al., 2020; Hakuba et al., 2021; Marti et al., 2022; Meyssignac et al., 2023; von Schuckmann et al., 2023).

Current estimates of the EEI are based on both measurements by satellite radiometers and inventories from ocean heat content measurements. For satellite radiometry of the EEI, the current flagship satellite mission is Clouds and the Earth's Radiant Energy System (CERES) (Wielicki et al., 1996), which uses scanning radiometers to measure the directional radiance at the satellite altitude, and then relies on angular dependence models to translate measurements into actual fluxes. These models can typically introduce corrections of the order of 10% for individual measurements (Loeb et al., 2018a).

For future measurements of the EEI, there are multiple potential methods that are being investigated and developed, ranging from extensions of current methods to more novel conceptual ideas. The former include the planned Libera mission (Hakuba et al., 2024), which is set to include scanning radiometers with higher spectral resolution than existing measurements, and requires new angular dependence models to be created (Gristey et al., 2023). Wide-field-of-view radiometers are another relatively established type of instrument that has seen renewed interest in recent years, concerning both instrument design and the potential for high-resolution measurements with large constellations with dozens of satellites (e.g. Gristey et al., 2017; Schifano et al., 2020; Swartz et al., 2019). By contrast, the more novel ideas include moon-based observation systems (Zhang et al., 2022, 2023). An alternative concept relies on the influence of radiation pressure on the motion of a satellite, and would ideally use spherical black satellites equipped with accelerometers and translate the acceleration into net flux measurements (Hakuba et al., 2023).

In this study, we specifically investigate the potential of simple wide-field-of-view radiometers. This is connected to the European Earth Climate Observatory (ECO) mission proposal, the general measurement concept of which is illustrated in Fig. 1: a constellation of satellites, each measuring the EEI by differential measurement of incoming and outgoing radiative fluxes, for an absolute uncertainty of the annual EEI within $1.0 \ \mathrm{Wm^{-2}}$. The main instruments are four identically designed wide-field-of-view radiometers that measure the radiation from limb to limb. Rotation of the satellite enables differential calibration between the Earth-facing and space-facing radiometers, so that certain common systematic biases can be eliminated from the final EEI measurement. Spare radiometers that are normally kept closed make it possible to monitor the drift of the main radiometers over time. In addition, cameras allow limited distinction between longwave and shortwave signals, and spatial resolution.

Here we focus only on sampling errors using the Earth-facing radiometer, with the intention of prioritising the accuracy of the long-term global mean over spatial and temporal resolution. In particular, we investigate the effects of diurnal and annual sampling issues in an otherwise idealised framework. There are many other potential sources of error. For example, an earlier study found that the dominating source of error for the ERBE satellite mission was the thermal environment of the instruments (Wong et al., 2018). Other studies have also analysed the requirements of a reference level for TOA radiation studies (Loeb et al., 2002). Neither of these types of errors is addressed in the current study. This current work considers purely Lambertian emission, but a detailed investigation into the effects of the angular dependence of the radiation is planned for a future study.

The diurnal cycle presents certain difficulties as a result of variations in the sampling of a single satellite, something that has been known since at least the 1970s (Campbell and Vonder Harr, 1978; Salby, 1988; Kirk-Davidoff et al., 2005; Taylor

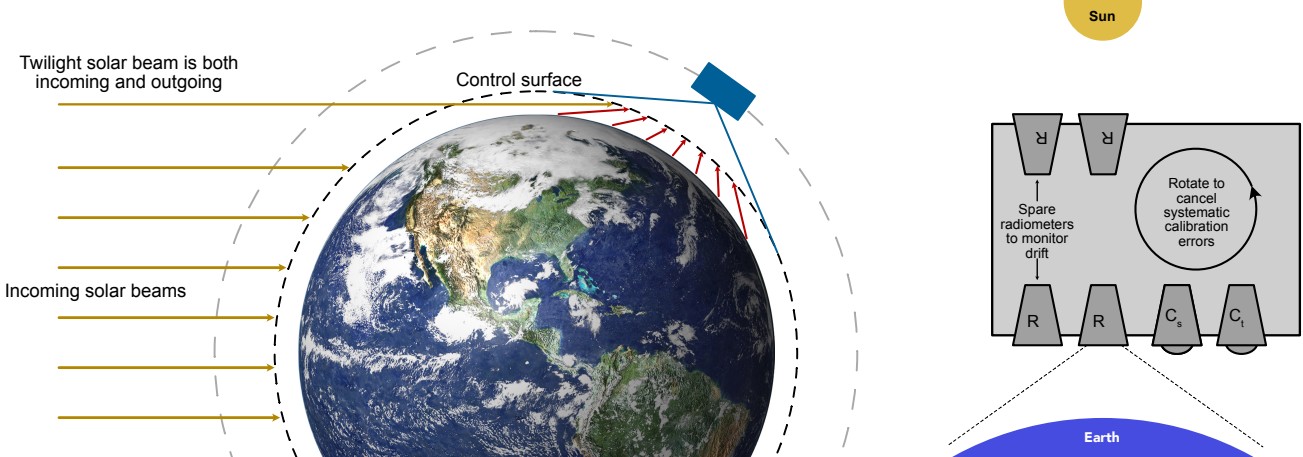

**Figure 1.** Measurement principle of the proposed ECO satellite mission. Left: schematic of the observation from each satellite, measuring all radiation from within the control surface. Right: simplified view of the intended satellite payload, namely the four wide-field-of-view radiometers (R) and the shortwave ($C_s$) and total ($C_t$) cameras that observe the Sun and the Earth. The idealised viewing geometry considered in the current study is presented in more detail in Sect. 2.3.

and Loeb, 2013). Several studies have investigated the potential of satellites that gradually change the observed local time, usually with a goal of monthly resolution in time and regional or higher spatial resolution (Campbell and Vonder Harr, 1978; Vonder Harr and Smith, 1979; Kirk-Davidoff et al., 2005; Smith et al., 2014; Gristey et al., 2017). Nevertheless, more recent efforts have typically considered sun-synchronous orbits and relied on a diurnal model to synthesise full sampling of the diurnal cycle, which requires that the model does not introduce additional errors (Young et al., 1998; Doelling et al., 2013). Our objective is to manage these issues without a diurnal model, instead relying on the direct sampling of the diurnal cycle and introducing as few a priori assumptions as possible in our synthetic measurement of the annual global mean. In principle, the difficulties could be mitigated by using a large number of satellites for dense sampling of the whole Earth at each moment in time. In practice, of course, any real mission will face physical, logistical and budget restrictions on the number and location of satellites. It is therefore interesting to explore how the sampling error of the EEI depends on the chosen orbits.

## 2 Methods

The overarching goal of the methods described below is to simulate what an idealised satellite would measure, in order to investigate satellite sampling issues. To that end, we use reference data for the radiation field and a measurement kernel to generate each individual measurement. This is combined with a framework for simulating satellite positions in space, and finally converting the measurement time series into global averages that can be compared with those of the original reference data.

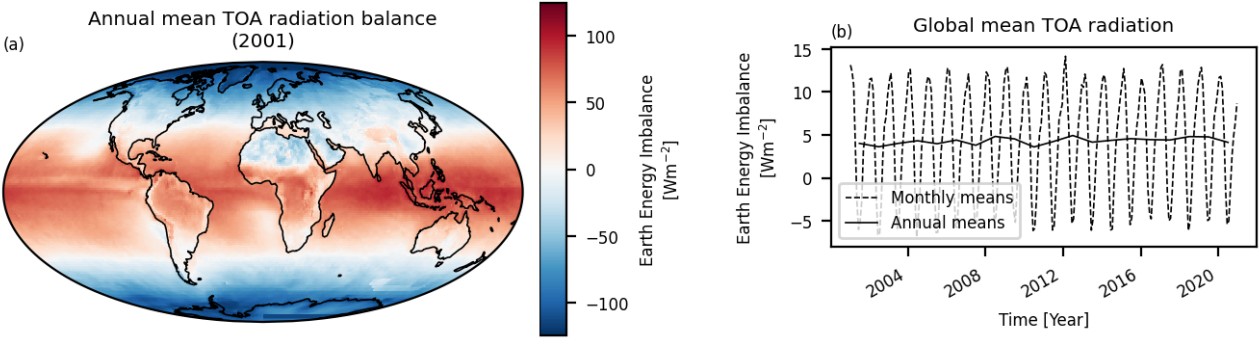

**Figure 2.** (a) Map of annual-mean net radiation. (b) Reference monthly (dashed) and annual (solid) mean time series of global mean Earth Energy Imbalance (EEI) at TOA. Based on 2001-2020 CERES SYN1DEG data.

## 2.1 TOA radiation from CERES reference data

As the best available reference for top-of-atmosphere (TOA) radiation, we used data from the CERES satellite mission. CERES offers two particularly relevant data products: EBAF (Energy Balanced and Filled) where the TOA net flux is constrained to match EEI estimates from ocean heat storage measurements, and SYN (synoptic) where focus is on regional and diurnal behaviour. We chose to use the SYN product (Edition 4A) in order to be able to investigate regional sampling issues and the impact of the diurnal cycle (NASA/LARC/SD/ASDC, 2017). By contrast, the particular absolute magnitude of the reference radiation budget is less important in this study since the results are all analysed relative to this reference magnitude. Specifically, we used all-sky TOA fluxes (hourly data on a $1° \times 1°$ grid).

The time-averaged global TOA radiation field shows a large net influx of energy at lower latitudes and a large net outflux of energy at higher latitudes, with some regional variation as a result of smaller-scale features (Fig. 2). In the global mean, the monthly EEI time series fluctuates between nominally -5 $\mathrm{Wm}^{-2}$ and +15 $\mathrm{Wm}^{-2}$, while the year-to-year variations in the annual time series are an order of magnitude smaller (Fig. 2). It is worth noting that this overall EEI time series is known to be inconsistent with current best estimates (Loeb et al., 2018b). This work nevertheless treats the SYN EEI value as if it were the known truth reference level, for the purpose of evaluating the performance of the hypothetical satellites. As such, the nominal 5 $\mathrm{Wm}^{-2}$ SYN bias does not affect the conclusions of the study.

We assume that the Earth is a perfectly spherical emitting shell with radius 6371 km, corresponding to TOA at zero altitude for simplicity, and radiative fluxes given by the all-sky TOA fluxes from the CERES reference data. This emission is assumed to be Lambertian. No atmospheric twilight transmission was included (Fig. 1, see also Sect. 2.3).

## 2.2 Measurement kernel

Mathematically, each individual measurement of the satellite-level radiative flux $F$ can be described by an integral over solid angle of the radiance from each visible surface element towards the satellite, projected onto the satellite normal (Smith and

Green, 1981; Green et al., 1990). Nevertheless, because we are using CERES data for radiant exitance (i.e. the non-directional total flux), instead of radiance, the integral is expressed in a slightly different way and performed over surface area. For a Lambertian surface, the radiance is independent of the viewing angle. If the total flux emitted by the surface element is $M$, the radiance $I$ from that same element along a zenith angle $\theta$ is $I = \frac{M\cos(\theta)}{\pi}$. Taking into account the $\frac{1}{d^2}$ decrease of the irradiance at the measurement location with distance $d$ and a perfect cosine instrument response, the overall integral is:

$$F = \int_A dA \frac{\cos(\theta_{SAT})\cos(\eta)}{\pi d^2} M, \tag{1}$$

where $A$ is the satellite-visible area of the emitting shell. The satellite viewing angle $\eta$ and zenith angle $\theta_{SAT}$ are shown in Fig. 5. For non-Lambertian emission, the integral would require an anisotropy factor, which is typically parameterised as a function of the relevant angles and grouped into an angular dependence model (ADM) for a given scene type, to account for the angular dependence of the radiation (Loeb et al., 2003, 2005). Note that this is separate from a shape factor, which would be used to invert the satellite-level measurement to an estimate of the flux at a different altitude (Green and Smith, 1991). For a non-perfect instrument response, the ideal $\cos(\eta)$ factor would no longer apply, and would in general be replaced by some instrument-specific function of $\eta$.

In this idealised Lambertian case, the measurement kernel is at a maximum immediately below the satellite, and decreases for surface elements further from the satellite subpoint. As a result, surface elements near the centre of the satellite footprint dominate the overall measurement, as illustrated by the footprint weights shown in Fig. 3. For instance, half of the signal originates from within six degrees of central angle. This also means that the overall measurement is more sensitive to deviations from the ideal $cos(\eta)$ factor in the instrument response at small $\eta$ than at large $\eta$. Small-$\eta$ deviations would typically be caused by inhomogeneities within the core of the instrument, while large-$\eta$ deviations may instead be the result of geometric effects at the limits of the field of view.

The radiation field observed at the satellite altitude is the result of a convolution of the TOA field and the measurement kernel. As illustrated in Fig. 4, the resulting field retains large-scale features such as general differences between the poles and the equatorial regions, but smaller-scale features are lost in the smoothing process.

## 2.3 Viewing geometry and instrumentation

The geometry of a satellite observing a given point can be illustrated by a triangle with corners at the satellite, the surface point and the centre of the Earth, as shown in Fig. 5.

The synthetic instrument being considered is an idealised wide-field-of-view radiometer, inspired by both first-principles studies and actual instrument designs (Mishchenko et al., 2016; Schifano et al., 2020). Such an instrument integrates incoming radiation from the entire footprint and from across the electromagnetic spectrum. To capture the correct radiation field, the field of view has to be sufficiently large to cover not only the visible Earth segment, but also the atmosphere up to the altitude of a hypothetical control surface that contains the emitting atmosphere. The satellite would also periodically receive solar twilight radiation that passes through this atmospheric layer directly (Fig. 1). In our idealised framework, however, we treat the surface

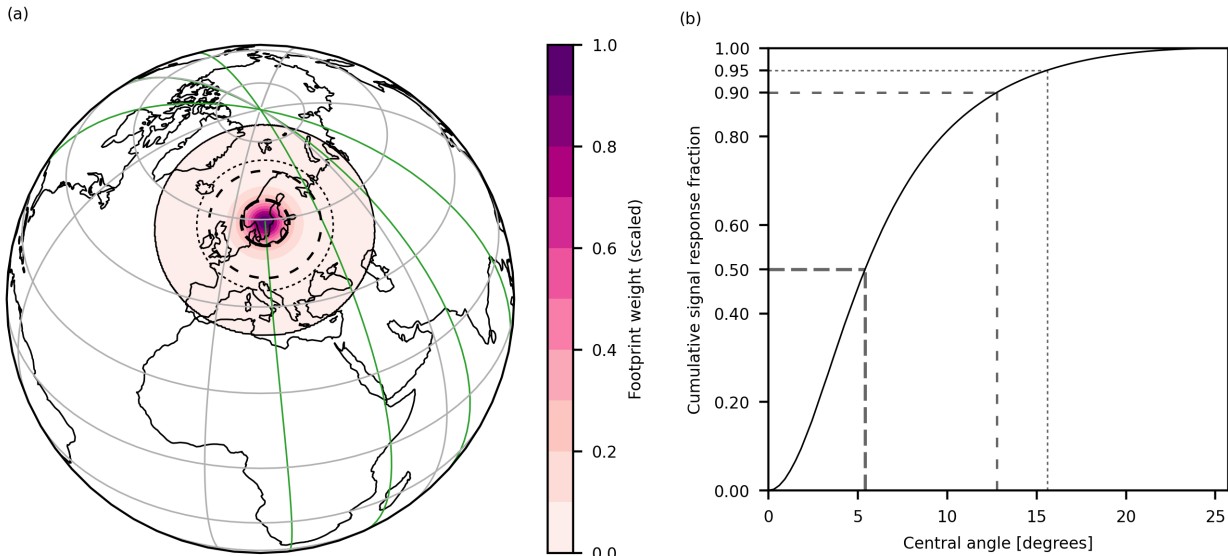

**Figure 3.** (a) Satellite path and sample footprint. The solid green line shows the satellite path of a polar satellite. The solid black line shows the edge of the satellite footprint for a satellite at 700 km altitude. The footprint weights are scaled by a common factor such that the maximum value (immediately below the satellite) is unity. (b) Cumulative signal response fraction as a function of central angle (see Fig. 5) for measurement of a homogeneous field. The dashed lines in both figures show the 50%, 90% and 95% thresholds for the cumulative response function. The maximum central angle is 25.71°.

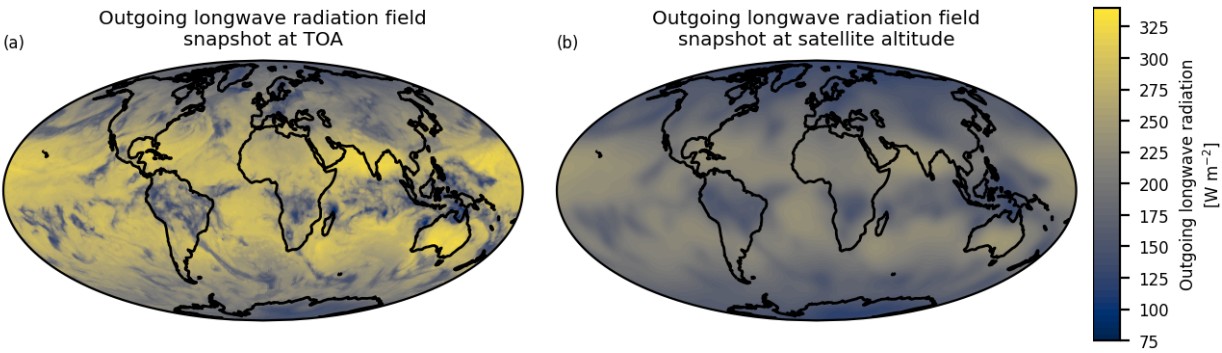

**Figure 4.** Sample 1-hour radiation field for outgoing longwave radiation at TOA [global mean 234.64 $\mathrm{Wm}^{-2}$] (a) and at the satellite altitude, based on Lambertian emission [global mean 190.48 $\mathrm{Wm}^{-2}$] (b). The radiation field at the satellite altitude, which is computed from (a) according to Equation 1, illustrates both the smoothing effect of the satellite footprint and the decreasing magnitude with the square of the fractional orbital radius (parameters defined in Fig. 5): $(R_E/R_{SAT})^2$.

of the Earth as an emitting shell that exactly fills the satellite field of view, and do not consider incoming solar radiation that may contribute to the outgoing radiation.

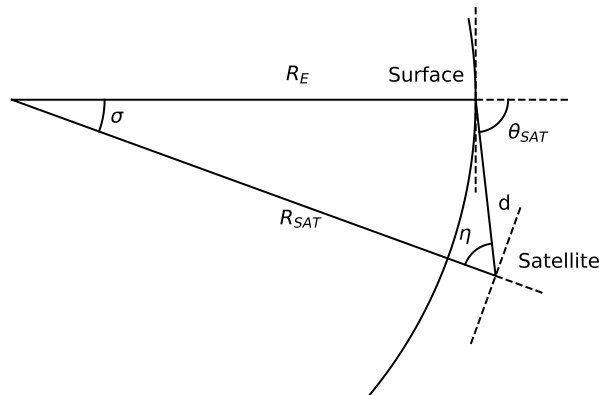

**Figure 5.** Diagram of viewing geometry for a satellite observing a point on the Earth's surface. The angles marked are the satellite zenith angle $\theta_{SAT}$, the satellite view angle $\eta$ and the central angle $\sigma$. Also marked are the distance between the satellite and the surface point $d$, and the distances from the centre of the Earth to the surface point and to the satellite $R_E$ and $R_{SAT}$. The dashed lines show the local normal and horizontal at the surface and the satellite.

The angular size of the real Earth varies slightly between the equator and the poles because of the equatorial bulge of the planet, but this has limited effect as long as the field of view allows the satellite to see from horizon to horizon. For a spherical Earth, the required field of view is constant at $2\sin^{-1}(R_E/R_{SAT})$, which for our chosen radius is 128.6°. We assume that this instrument makes one measurement every minute, with zero satellite pointing error and with a perfect cosine response to accurately measure the irradiance at the satellite altitude.

## 2.4 Satellite orbits

To generate synthetic satellite measurements, idealised satellites were simulated at an orbital altitude of 700 km. They were initialised in circular zero-drag orbits with four different inclinations, as illustrated in Fig. 6. Because the orbital motion is fundamentally caused by the gravitational force from the Earth, gravitational variations due to the aspherical shape of the Earth cause a torque on a satellite in a non-polar orbit, which causes the orbital plane to precess, i.e. the satellite orbital plane will rotate over time in a celestial reference frame (Kaplan, 2006). For a first-order expansion of the spherical harmonics of the geopotential, the rate of precession is proportional to the cosine of the inclination (IERS Convention Centre, 2010; Rees, 2012). As shown in Fig. 6, the inclination also determines the maximum latitude that the satellite reaches, which may lead to blank spots in the sampling.

First consider a polar satellite, i.e. with a 90° inclination. By definition, such a satellite passes directly over the poles, with no net torque on the satellite and hence no precession. The orbital plane thus has a constant orientation in the celestial reference frame, and the polar satellite gradually observes different local solar times as the Earth orbits around the Sun. Provided that

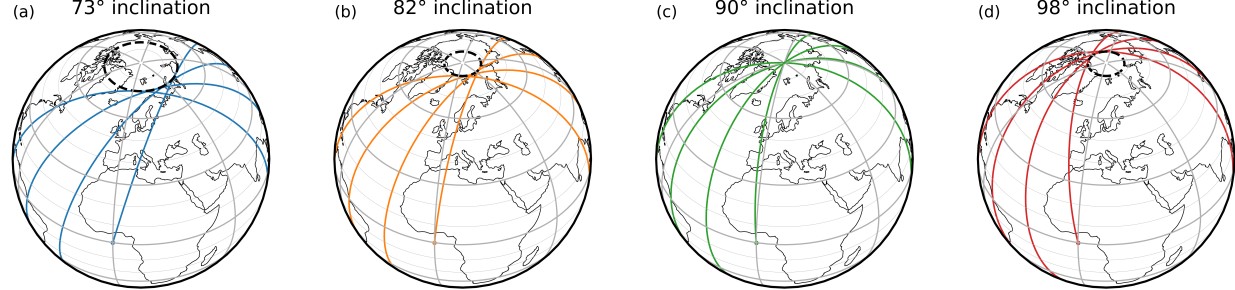

**Figure 6.** Satellite orbits with different inclinations. From left to right: 73°, 82°, 90°(polar), 98°(sun-synchronous). The inclination describes the angle the satellite makes against the equator as it crosses from the southern hemisphere to the northern hemisphere, which is the same as the maximum latitude it reaches before returning south.

the satellite is observing on both sides of the planet in one orbit (i.e. both night and day, or dusk and dawn), one such satellite samples the diurnal cycle in half a year.

Next consider a satellite in a sun-synchronous orbit: the rate of precession exactly matches the orbital rate of the Earth around the Sun, so that the satellite always observes the same local time at the equator, with gradually larger deviations from the equator local time as the satellite observes higher and higher latitudes. As a result, a sun-synchronous satellite always samples the same part of the diurnal cycle. For our chosen orbital altitude, this corresponds to an inclination of 98.1716°, which is labelled as 98° in the rest of this article. Technically, this is a specific case of a precessing satellite, but in this article it will only be referred to as sun-synchronous.

Lastly, consider a satellite where the precession rate is such that the diurnal cycle is sampled multiple times per year. With our chosen orbital altitude, orbits with inclinations of 81.81° and 73.45° (labelled as 82° and 73° in the rest of this article) sample the diurnal cycle four and six times per year. These two orbits are described as precessing in this article. In principle, it is possible to use precessing orbits with even lower inclinations, but a simplified analysis indicated that the benefit decreased below approximately 60°.

For our simulations, the orbital trajectories were computed using the SGP4 simplified perturbation model (Vallado et al., 2006), with the settings and parameters shown in Table 1. The effect of satellite thrusters for orbital maintenance was included by setting the drag coefficient to zero.

## 2.5 Conversion to global mean

In order to compute a meaningful global average from the series of individual measurements, it is important that the averaging method accounts for the original sampling used when producing the measurements. For example, it is straightforward to see that a polar satellite will sample more densely at higher latitudes than at the equator, because of the geometry of the Earth. On a global scale, then, each individual measurement near the pole should arguably carry less overall weight than each individual

| SGP4 Parameter name | Value |
|---|---|
| Gravity model (identifier for a collection of preset geopotential constants) | WGS72 |
| Mode (legacy mode or modern improved mode) | i (improved) |
| Epoch (reference time when parameters are specified) | 2001-01-04 08:52:00 (Earth at perihelion, i.e. closest to the Sun) |
| Bstar (drag coefficient) | 0 |
| Eccentricity (elongation of orbit ellipse) | 0 |
| Argument of perigee (ellipse orientation in orbital plane) | 0° |
| Inclination (tilt of the orbital plane relative to the Earth equatorial plane) | 73.45°, 81.81°, 90.0°, 98.1716° |
| Mean anomaly (angular position along orbit at epoch) | 0° |
| Mean motion (angular velocity of orbit) | 0.0637 radians/minute[*] |
| Right ascension of ascending node (orientation of orbital plane at epoch, relative to Earth equatorial plane[†]) | 0°, 15°, 30°, ..., 165° |

**Table 1.** Settings used to compute satellite trajectories with the SGP4 simplified perturbation model (Vallado et al., 2006; Vallado and Crawford, 2008). [*]This is only an approximation. The exact value of the mean motion is determined from the orbital period, and is ultimately computed as $\sqrt{GM_E/a^3}$, where $G = 6.6743 \cdot 10^{-11} m^3 kg^{-1} s^{-2}$ is the gravitational constant, $M_E = 5.97237 \cdot 10^{24} kg$ is the mass of the Earth and $a = 7.071 \cdot 10^6 m$ is the radius of the orbit. [†]Specifically, the angle is given with the March equinox as the reference.

measurement near the equator. For a polar satellite, this is easily addressed by a sinusoidal weighting with measurement latitude, but in order to also handle non-polar satellites, we bin the measurements on a coarse latitude/longitude grid and process each bin separately. It is important that each bin be big enough to contain sufficient values to produce an accurate mean, while small enough to minimise biases within the bin and biases from the subsequent global averaging process. Given that the satellites can move almost 4° in latitude between measurements, the latitude bin width should be at least this size to avoid missing bins even when passing over them. We have investigated the effect of different bin sizes from 1° × 1° to 30° × 30°, and find that the trade-off between bin return frequency and global-averaging error results in minimal errors for bin widths of nominally 3.6° to 10°. Figure 7 illustrates that the shortwave results show overall positive biases from the individual satellites and negative biases from the grids, while the longwave results show the opposite. These errors partially compensate each other in the total results, but the shortwave biases still dominate.

For this study, a 5° by 5° grid was used. This was chosen as it is in the previously mentioned range for minimal bin-size
errors and also means that each individual grid cell typically is observed at least every 14 days, with most return times being
far shorter. On average, this corresponds to between 200 and 250 measurements per bin and year, but the measurements for
each bin are not evenly spaced in time as a result of the satellite orbit. The overall distribution of return times is shown in Fig.
8. Regardless of inclination, 80-90 % of measurements occur with a return time below 48 hours, approximately evenly split
between the first and second 24-hour periods. The few remaining return times are spread across the tail of the distribution,
with maximum return times of 9 days (90°), 15 days (73°) or 23 days (82°). A more detailed zonal distribution of the return
times is shown in Fig. 9. There are some periodic zonal patterns to be seen in the occurrences of the long return times, but all
latitudes nevertheless have typical return times below two days, in line with the previously mentioned overall distributions. As
a consequence of the orbital trajectory, precessing orbits notably result in a greater number of measurements near the minimum
and maximum orbit latitudes, with shorter return times in these specific regions as a result.

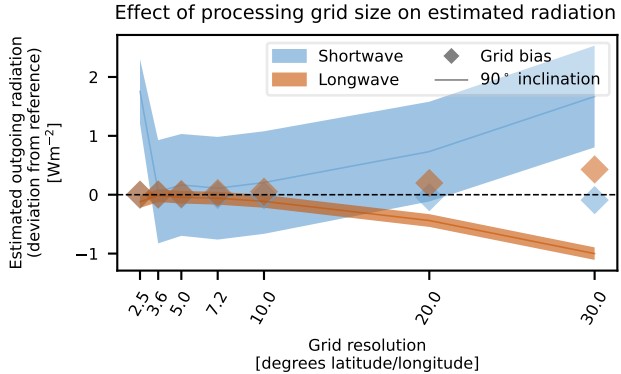

**Figure 7.** Estimated outgoing shortwave and longwave radiation and the effect of the processing grid size (see Sect. 2.5), for the period
2001-2020. The shortwave component includes the correction described in Eq. 2 near the end of Sect. 2.5. The lines mark the annual mean
deviation from the reference truth for results from individual 90° satellites, with shaded regions showing the annual standard deviation. The
same set of synthetic satellite measurements were used for all cases, but they were processed on grids with different spatial resolutions. The
diamonds mark the bias associated with remapping the true reference field onto the grid in question.

A less obvious consideration is the variation in diurnal solar irradiation over the course of a year, where the local diurnal
cycle is modulated by the amplitude of the overall incident solar radiation, which depends on the distance to the Sun. Depending
on the orientation of the satellite orbital plane in relation to the orbit of the Earth around the Sun, this can lead to systematic
biases depending on how the resulting apparent diurnal profile compares to the true diurnal profile. To address this issue, we
apply a simple shortwave correction based on incoming and outgoing radiation, as described below. Note that this correction
requires a method to separate the full-spectrum radiometer measurements of outgoing radiation into longwave and shortwave
components, such as the proposed ECO cameras (Fig. 1). We mainly assume that the fraction of shortwave to total outgoing
radiation, and by extension the outgoing shortwave radiation itself, is measured perfectly, but we also perform a sensitivity test

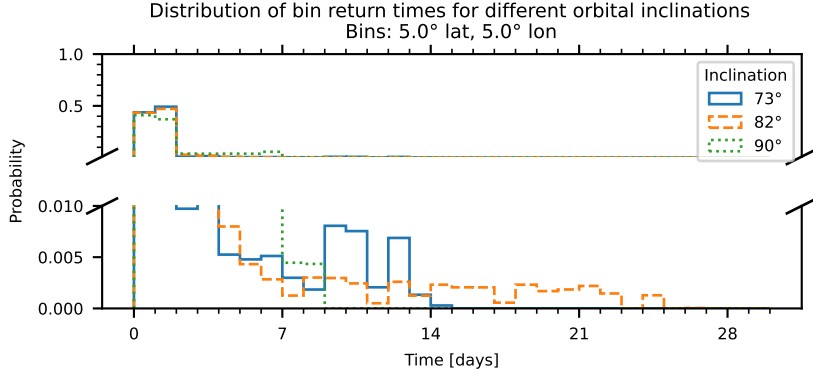

**Figure 8.** Probability density function of return time between subsequent measurements in each latitude/longitude bin, for different satellite inclinations. These are the results of a single satellite for 20 years.

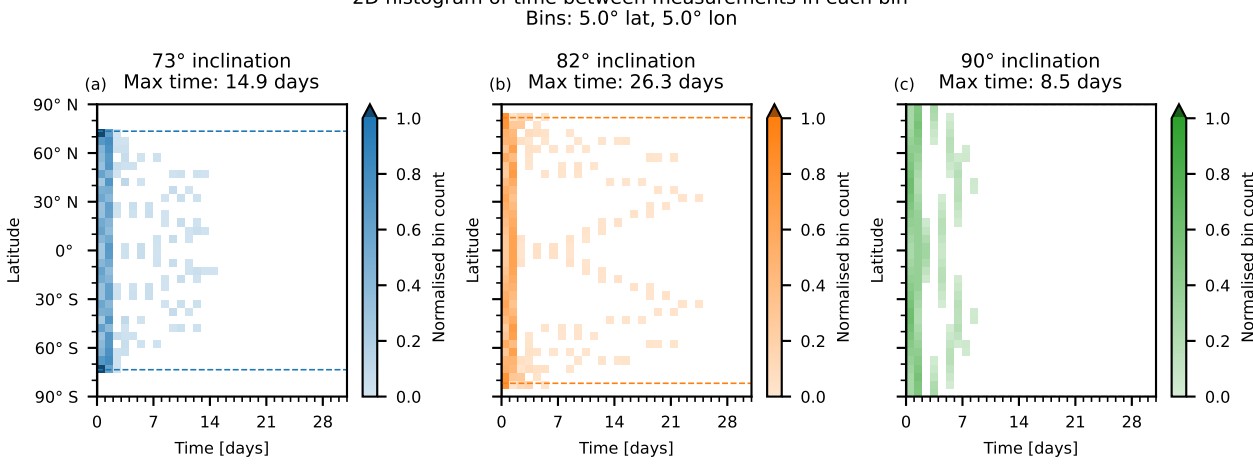

**Figure 9.** Distribution of return times between subsequent measurements in a single latitude/longitude bin, for different satellite inclinations. The results are for a single satellite over 20 years. The dashed lines show the latitude extent of the satellite orbit. The bin counts are normalised so that unity corresponds to the overall expectation value per latitude band, i.e. the total number of counts divided by the number of latitude bins. For non-polar orbits, the bins closest to the poles will have zero counts by construction, so the normalised count in the remaining latitude bins may exceed unity.

where this fraction is systematically $\pm 10\%$ different compared with the true value. We consider the incoming solar radiation to be much more predictable than this, and associated correction errors to be negligible by comparison.

The general algorithmic sequence of operations is shown schematically in Fig. 10. In more detail, these instructions were followed:

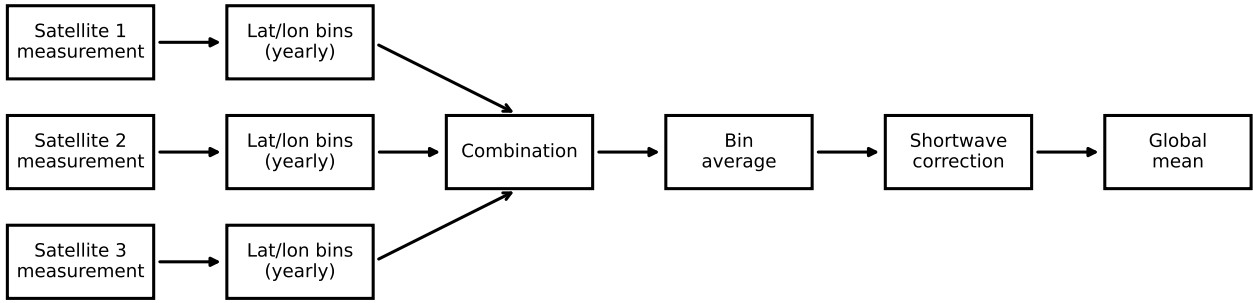

**Figure 10.** Schematic overview of the sequence of processing steps to convert satellite measurements into global mean values.

1. Compute synthetic measurement time series for each satellite with the kernel described in Sect. 2.2. Measurements are computed separately for outgoing longwave radiation (OLR), outgoing shortwave radiation (OSR) and incoming shortwave radiation (ISR), based on the corresponding fields in the reference CERES data.

2. Bin the measurements for each year on a 5° by 5° grid, based on the coordinates of the satellite subpoint at the time of measurement. Store the annual sum of measurement values in each bin.

3. Combine binned measurements from the different satellites within the chosen constellation as a sum of the corresponding measurement values. Note that for some constellations, a given satellite may only contribute to specific latitude bands (Sec 3.3). If only a single satellite is being used, this step has no effect.

4. Within each bin, compute the average measurement value for each radiation variable.

5. Compute a corrected value for OSR, based on the measured ISR value, $ISR$, and the corresponding perfectly sampled reference truth ISR value from CERES, $ISR_{CERES}$. For each bin $b$ and year $y$:

$$OSR_{corrected}(b,y) = \frac{ISR_{CERES}(b,y)}{ISR(b,y)} OSR(b,y) \tag{2}$$

As shown in Fig. 11, this shortwave correction reduces the magnitude of the deviation from the reference. For conve-
nience, the orbital plane is parametrised according to the equivalent local time observed at the reference epoch (Table 1).

6. Compute annual global means of the average binned measurements, weighted by the bin areas.

The resulting global annual mean EEI can then finally be computed as $ISR - (OLR + OSR_{corrected})$.

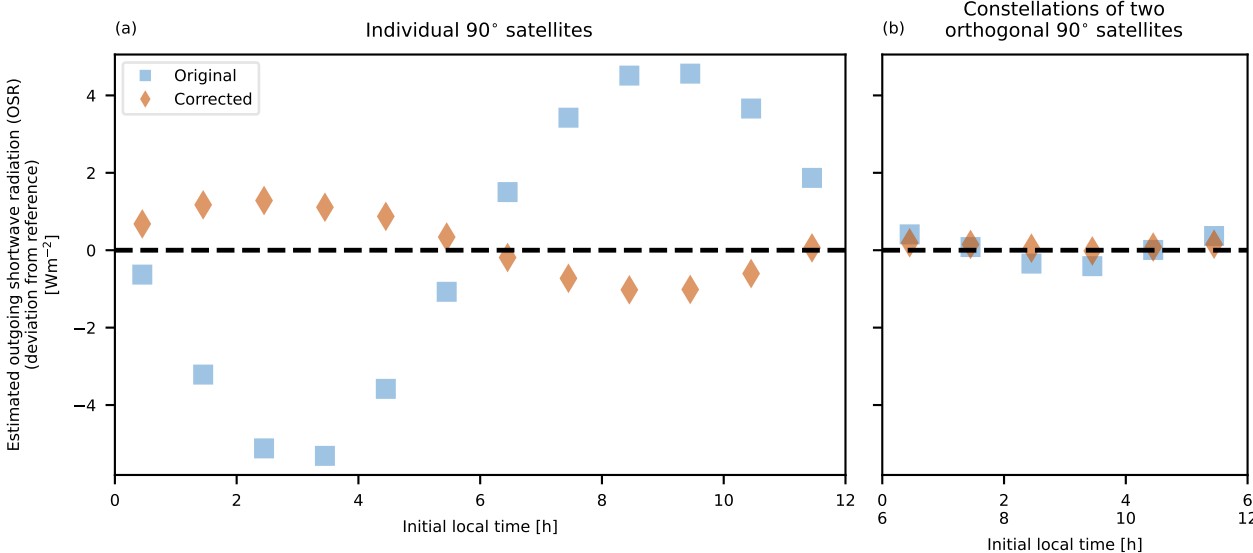

**Figure 11.** Effect of correction of estimated outgoing shortwave radiation, based on measurements from twelve polar satellites for the year 2001. Panel (a) shows the deviation in estimated outgoing shortwave radiation using measurements from individual 90° satellites, with and without the shortwave correction described in Sect. 2.5. Panel (b) shows corresponding results for constellations of two orthogonal satellites, combined as per Sect. 2.5. The mechanics of the sinusoidal single-satellite deviation are addressed in Sect. 3.2. Constellations of polar satellites are detailed further in Sect. 3.3 and illustrated in Fig. 17.

## 3   Results

With the above explanation of how to translate measurement series into a global mean value, all that remains is to apply this method to specific satellite orbits and constellations. We shall see that different types of orbits, and combinations thereof, have qualitatively different performances regarding the EEI estimate. Let us start by considering sun-synchronous satellites.

### 3.1   Sun-synchronous satellites

By definition, the diurnal sampling of individual sun-synchronous satellites is limited to two samples per day at fixed times for a
given latitude. As such, we can expect a straightforward mean of the measurements from a single satellite to be systematically biased depending on the observed local solar time. This is shown in Fig. 12 for the estimated total outgoing radiation. Of course, an estimate of the actual quantity of interest, the EEI, would in principle also require a measurement of the incoming component. Nevertheless, due to the fact that the sun-synchronous satellite by construction has a near-constant viewing angle to the Sun, we can expect that measurements of the incoming solar radiation would be very stable. Hence the main quantity

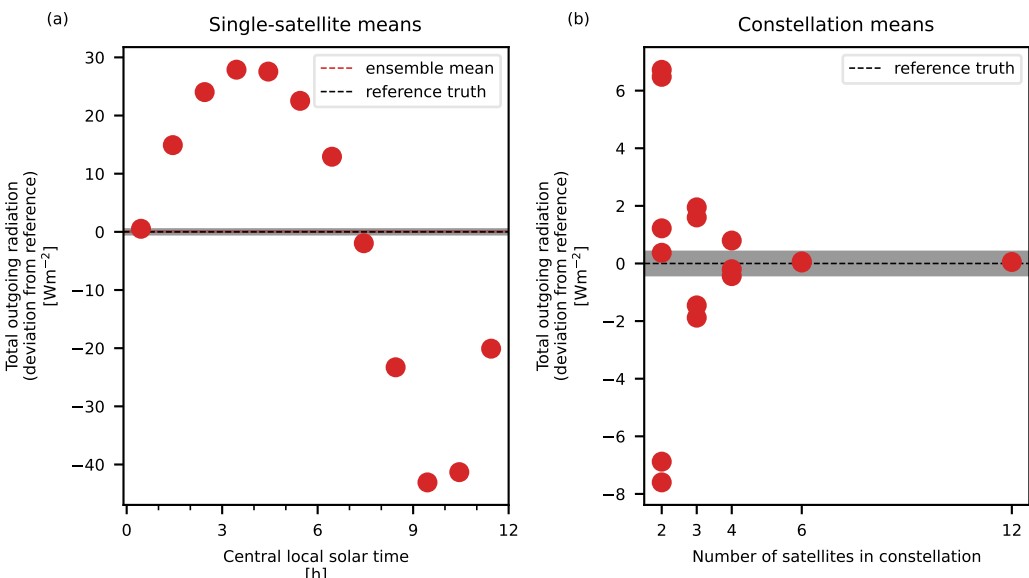

**Figure 12.** (a) Observed total outgoing radiation for sun-synchronous satellites at different local solar times. (b) Mean total outgoing radiation for constellations of evenly spaced sun-synchronous satellites. These values do not use albedo-corrected shortwave radiation, as the limited diurnal sampling of the sun-synchronous satellites actually means that the albedo correction makes the estimate worse. The error bars and shaded areas show the standard deviation for the annual means.

of interest and the main source of uncertainty is the outgoing radiation component. The biases of up to several tens of $\mathrm{Wm}^{-2}$ mean that a single sun-synchronous satellite is insufficient for measurement of the EEI in this way.

Can better performance be achieved by combining multiple sun-synchronous satellites in a constellation? The more individual satellites there are, the better the diurnal sampling, and six satellites evenly spaced in sampling local time are enough for an overall uncertainty well below $1 \ \mathrm{Wm}^{-2}$ in the outgoing radiation (Fig. 12). Sun-synchronous constellations are briefly

discussed further in Sect. 3.3. However, six may be an unfeasibly large number of satellites for a real mission. If fewer sun-synchronous satellites are to be used, they would require a model of the diurnal cycle to compensate for the bias, as used in CERES products (Doelling et al., 2013). Because we want to limit our usage of such models, so as not to introduce errors, we might instead consider the use of satellites that directly sample the diurnal cycle.

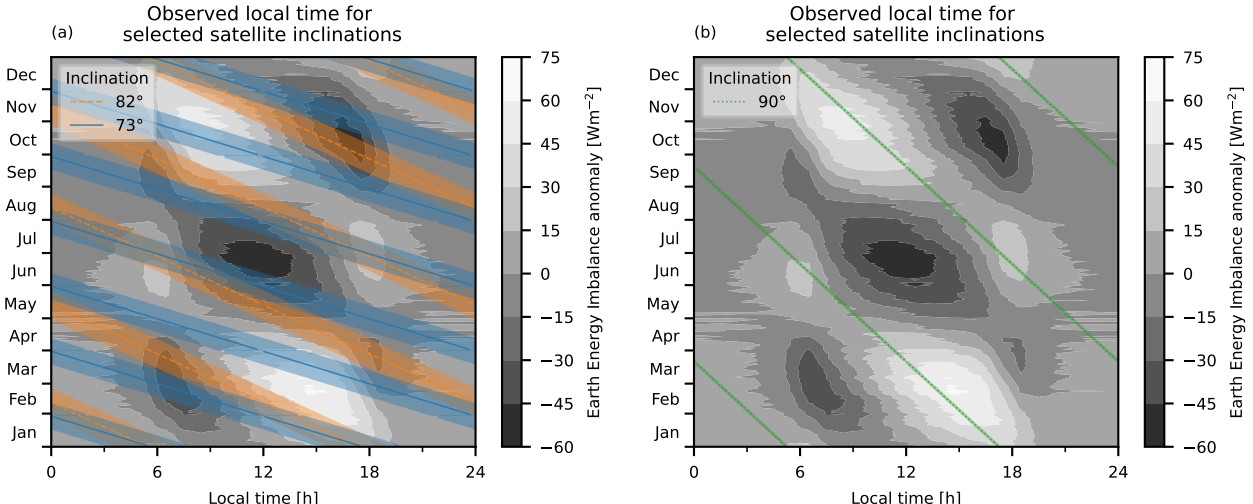

**Figure 13.** Plot of the variation in local solar time over the course of 2001 for satellites with different inclinations ((a) 73° and 82°, (b) 90°), showing both the daily median local time and the daily standard deviation of the local time. Underneath the local-time trajectories are contours of the local-time reference truth EEI anomalies, relative to the annual mean value for each local time.

### 3.2 Single satellites: polar & precessing

Unlike sun-synchronous satellites, both polar and precessing satellites gradually observe different local times over the course of a year. As mentioned in Sect. 2.5, this can potentially lead to systematic biases depending on the chosen satellite orbital plane. Three examples of satellite trajectories in local-time space are shown in Fig. 13. We see that the period of the polar orbit (90°, right panel) coincides with the period of the global variations of TOA imbalance, which in turn is dominated by the incoming solar radiation, such that a given polar satellite may consistently observe local maxima or minima and ultimately provide a biased estimate of the EEI (Fig. 14). By contrast, the precessing orbits have periods that are sufficiently different from the main period of the underlying EEI, that they achieve negligible sampling error from the annual and diurnal cycles (Fig. 14).

The spread in observed values can also be illustrated as a range of the latitude profile for each type of orbital inclination, as shown in Fig. 15. Individual polar satellites have the advantage that they cover the whole Earth, but they barely achieve an uncertainty below 1 $\mathrm{Wm}^{-2}$, and these large systematic errors need to be addressed. Lower-latitude satellites (73° and 82°) have much smaller biases in the order of only 0.1 $\mathrm{Wm}^{-2}$ or less, but on the other hand miss data from the polar caps.

We can conclude that no single satellite provides sufficient coverage of both the whole Earth and the diurnal cycle. A natural next step is to combine different kinds of satellites to mitigate these limitations.

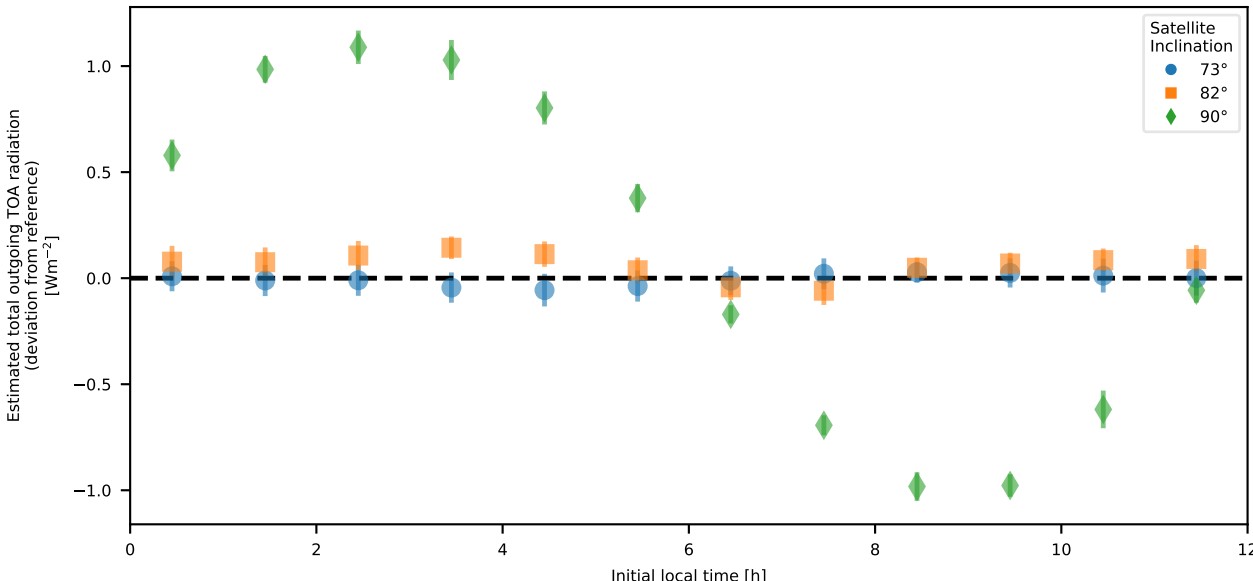

**Figure 14.** Total outgoing radiation as estimated by individual satellites with different orbital inclinations. Each point represents the mean annual deviation from the reference truth over the period 2001-2020, with error bars showing the standard deviation of the annual deviation. The reference truth is computed separately for each satellite, covering only the latitude range of the satellite. I.e. the 90° estimates are compared with the full global mean, whereas e.g. the 82° estimates are compared with an area-weighted mean of the true latitude profile from 82° S to 82° N.

## 3.3 Satellite constellations

Since the precessing 73° and 82° satellites were observed to achieve good diurnal sampling, one precessing satellite is used as a starting point in this section, and a 90° satellite is used to fill in only the otherwise missing data at the poles. Results for these combinations are shown in Table 2. The combined errors for the 73°+90° constellation are smaller than those of the 82°+90° constellation, so we can conclude that the higher diurnal sampling rate of the 73° satellite is more beneficial than the greater latitude range of the 82° satellite.

It is worth noting that the orientation of the orbital plane, which is a function of initial local time, for either satellite in the constellation makes only a very small difference in the final estimate. Essentially, the precessing satellites sample the diurnal cycle frequently enough for any initial differences amongst them to soon shrink. The polar satellites are effectively only measuring near the poles, where they would be measuring much the same radiation field regardless of their orbital plane, because all polar orbits converge at the poles. The latitude deviation profiles of these constellations are shown in Fig. 16.

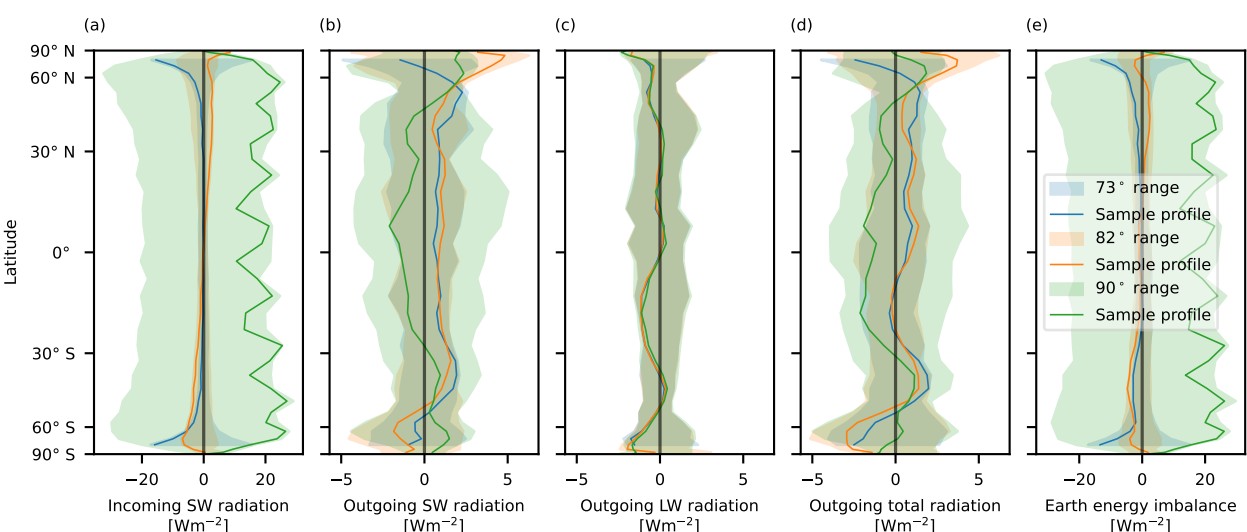

**Figure 15.** Deviation of TOA profile relative to smoothed reference truth, for single-satellite latitude profiles. The shaded regions show the minimum and maximum for the given inclination over 20 years, for all initial local times. The variations in the incoming shortwave radiation (a) and in the Earth energy imbalance (e) are dominated by the the effect of the initial local time. The variation in the outgoing longwave radiation (c) is dominated by year-to-year variations. The variations in the outgoing shortwave (b) and total radiation (d) are the result of both initial local time and year-to-year variations. The solid lines show the results for one satellite and one year, to indicate a typical latitude profile. The mean values for the outgoing total radiation are found in Fig. 14. The 73° and 82° satellites generally have lower deviations than the 90° satellites at lower latitudes, but on the other hand they miss data from the regions closest to the poles.

| Satellite inclination | 73°+90° | 82°+90° | 2x 90° | 3x 90° | 4x 90° | 73° | 82° | 90° |
|---|---|---|---|---|---|---|---|---|
| Ensemble size | 144 | 144 | 6 | 4 | 3 | 12 | 12 | 12 |
| RMS error [no SW corr] $(\mathrm{Wm}^{-2})$ | 0.45 | 0.23 | 0.32 | 0.07 | 0.04 | 0.49* | 0.23* | 3.56 |
| RMS error [SW corr] $(\mathrm{Wm}^{-2})$ | 0.08 | 0.10 | 0.10 | 0.04 | 0.04 | 0.08* | 0.10* | 0.78 |
| RMS error [SW corr, -10% OSR fraction bias] $(\mathrm{Wm}^{-2})$ | 0.08 | 0.11 | 0.11 | 0.04 | 0.03 | 0.08* | 0.11* | 0.37 |
| RMS error [SW corr, +10% OSR fraction bias] $(\mathrm{Wm}^{-2})$ | 0.10 | 0.10 | 0.09 | 0.04 | 0.05 | 0.10* | 0.10* | 1.20 |

**Table 2.** Root mean square (RMS) annual error in total outgoing TOA radiation, with and without shortwave correction (SW corr), relative to the reference truth. These errors are computed over all 20 years and all ensemble members. The values are based on measurements with different constellations: one precessing satellite (73° or 82°) in combination with one polar satellite (90°) to fill in data for the poles, or two, three or four polar satellites. Results are also shown for the case of systematic positive or negative bias in the camera-determined OSR fraction (Sect. 2.5). The results for individual satellites (73°, 82°, 90°), based on the data from Fig. 14, are included for comparison. *These results only cover the latitude range of the satellites, and thus do not include the full polar regions. The effect of missing polar data is discussed in Sect. 3.4.

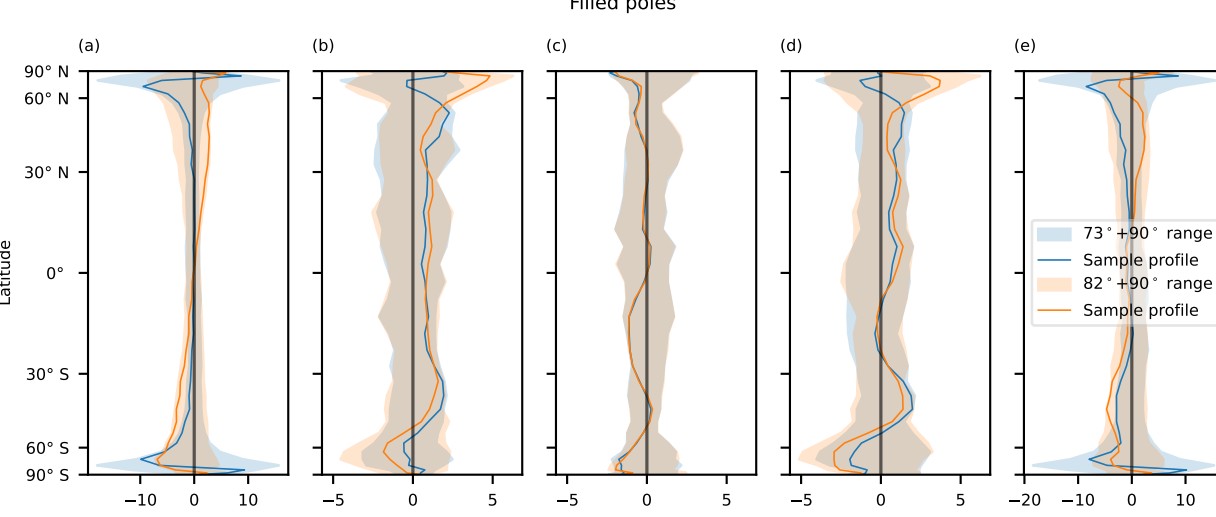

**Figure 16.** As Fig. 15 for constellations with filled poles.

| Satellite inclination | 2x 98° | 3x 98° | 4x 98° | 6x 98° |
|---|---|---|---|---|
| Ensemble size | 6 | 4 | 3 | 2 |
| RMS error [no shortwave correction] $(\mathrm{Wm}^{-2})$ | 5.69* | 1.74* | 0.56* | 0.16* |
| RMS error [with shortwave correction] $(\mathrm{Wm}^{-2})$ | 1.24* | 0.63* | 0.26* | 0.07* |
| RMS error [SW corr, -10% OSR fraction bias] $(\mathrm{Wm}^{-2})$ | 1.68* | 0.74* | 0.29* | 0.06* |
| RMS error [SW corr, +10% OSR fraction bias] $(\mathrm{Wm}^{-2})$ | 0.83* | 0.52* | 0.24* | 0.08* |

**Table 3.** As Table 2 for constellations of sun-synchronous satellites (98°).

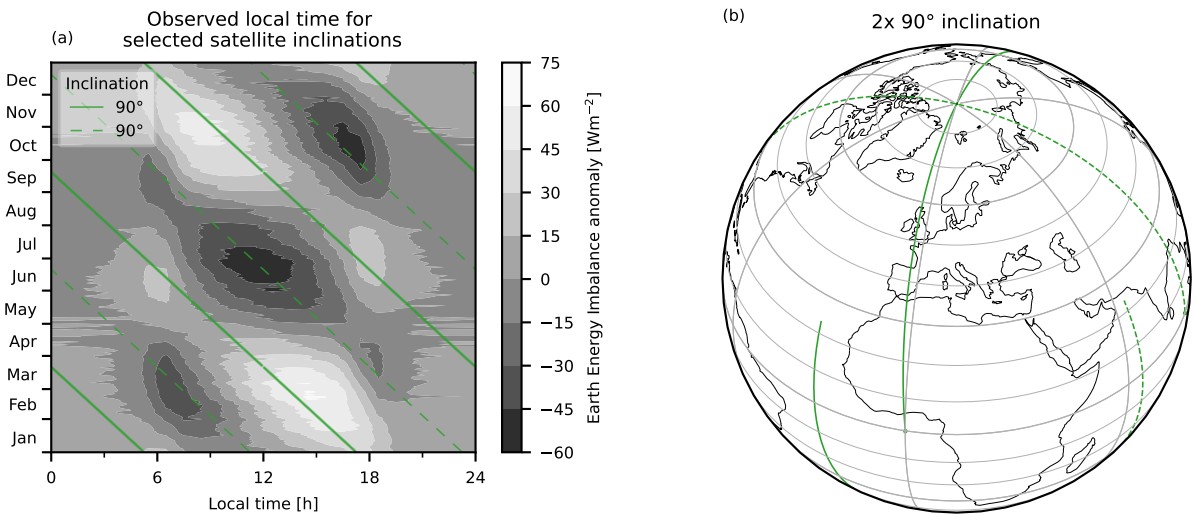

**Figure 17.** Left: As Fig. 13 for two orthogonal 90° satellites. Right: Sample trajectories for two orthogonal 90° satellites. As shown in the left panel, the complementary trajectories of the two satellites are such that they together observe both maxima and minima, and together reach a mean value that is much closer to the true mean than the mean from either satellite individually.

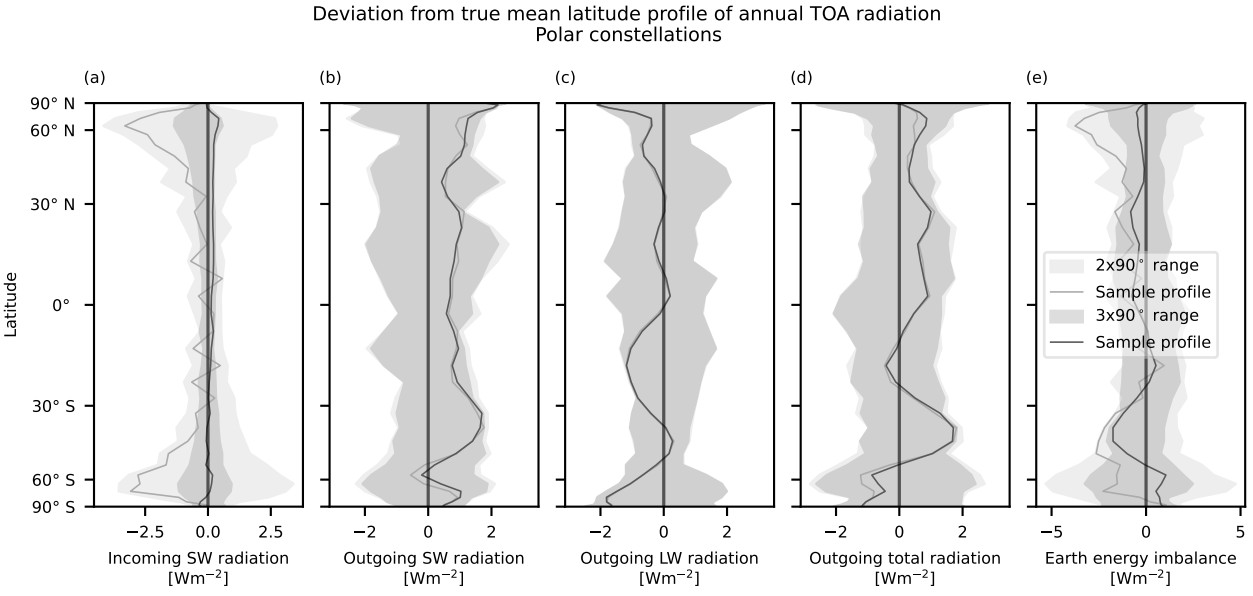

**Figure 18.** As Fig. 15 for constellations with multiple polar satellites.

Next, instead of using two different kinds of orbits to each shore up the weakness of the other, we consider the use of multiple polar satellites in combination to mitigate the systematic biases in their diurnal sampling. With two polar satellites, the two orbital planes should be orthogonal to each other, or equivalently six hours apart in observed local time (e.g. 00:00/12:00 and 06:00/18:00). The trajectories and local-time paths of such a constellation are illustrated in Fig. 17. In a similar way, a constellation with three polar satellites should observe local solar time four hours apart (e.g. 00:00/12:00, 04:00/16:00 and 08:00/20:00) in order to achieve an even sampling of the diurnal cycle, and a constellation with four polar satellites should have the orbital planes three hours apart.

As shown in Table 2, the triple-polar constellation achieves a lower error, with the RMS error 0.04 $\mathrm{Wm}^{-2}$ well within 0.1 $\mathrm{Wm}^{-2}$. The double-polar constellation RMS error of 0.10 $\mathrm{Wm}^{-2}$ is slightly larger, exactly at the 0.1 $\mathrm{Wm}^{-2}$ threshold. The latitude deviation profiles of these constellations are shown in Fig. 18. The double polar constellation has comparable performance to a 73°+90° or 82°+90° constellation. The inclusion of a third satellite can lead to further improvement, as the triple polar constellation performs noticeably better than all three two-satellite constellations. The shortwave correction consistently leads to a clear improvement for all these constellations, with only a small sensitivity to a 10% bias in the OSR fraction. The four-polar constellation performs only marginally better than the triple-polar constellation, with the same shortwave-corrected RMS error, but it is worth noting that the fourth satellite seemingly negates the need for the shortwave correction altogether.

Lastly, we briefly consider the performance of these constellations compared with constellations of multiple sun-synchronous satellites. As was mentioned in Sect. 3.1, single sun-synchronous satellites result in too large uncertainties with the current methodology, and multiple satellites are required to achieve the desired level of uncertainty. Table 3 shows this in more detail, and illustrates that six sun-synchronous satellites are necessary for results comparable to two or three satellites in one of the configurations from Table 2. The sun-synchronous satellites could still have other, practical benefits related to e.g. intercalibration and shared launches since there are many other satellites in sun-synchronous orbits, but it is hard to include this in the current assessment in a meaningful way.

To summarise the performance of the different constellations, we note that a sun-synchronous constellation requires a large number of satellites to be reliable, but all of the previously mentioned combinations of two or more 73°, 82° or 90° satellites result in typical uncertainties within 0.1 $\mathrm{Wm}^{-2}$. The results also indicate that the shortwave correction introduced in Sect. 2.5 is robust even for biases of $\pm 10\%$ in the camera-measured shortwave fraction, but the improvements from the shortwave correction can otherwise be achieved by including additional satellites.

### 3.4 Single satellites: revisited

In previous sections it has been demonstrated that constellations perform better than individual satellites, as we might intuitively expect; nevertheless, it is worth briefly revisiting the single satellites to further examine their potential. For instance, there may be logistical constraints such that only one satellite can be used for a certain period, or the permanent loss of a satellite may require the processing method of the remaining data to be adjusted.

In section 3.2 we identified two weaknesses of individual satellites: missing data from the poles (73° and 82° satellites) and biased sampling of the diurnal cycle (90° satellites). In order to address the former, these satellites could potentially have the

missing data filled in. This might be done by adapting measurements from the ECO cameras, which would still observe all latitudes even if the satellite itself only reaches 73° S/73° N. However, radiances from the polar regions would only be observed at relatively large satellite zenith angles, and would need an angular dependence model to translate these measurements into fluxes for these regions. Rather than consider the complex effects of all the potential components of such a model, let us instead focus on different levels of overall performance.

If we first neglect the missing data altogether, and compare the area-weighted mean of the satellite results directly with the reference global mean, it turns out that these results actually fall very close to the reference truth, with typical overall deviations in the order of 0.1 $\mathrm{Wm}^{-2}$ for the annual outgoing radiation. This is not so surprising, given that the 73° and 82° satellites already pass directly over most of the Earth's surface (95.9% and 99.0%, respectively). However, this relies on an unspecified implicit correlation between the missing polar regions and the measured rest of the world, instead of an explicit estimate of the missing data.

As a next step, let us assume that the chosen method somehow manages to perfectly fill the data gaps, which would mean that the 73° and 82° results from Fig. 14 and Table 2 would hold for the global mean. At an overall error of only approximately 0.1 $\mathrm{Wm}^{-2}$, this would be a very good result for a single satellite measuring the whole Earth. More realistically, the method for filling the gaps is likely to introduce additional errors, that may result in a systematic bias of the final result. If these missing data are filled with data that have a systematic bias of, say, 10 $\mathrm{Wm}^{-2}$ (nominally 3% of the global mean outgoing flux), this would then affect the final global average proportionally to the fractional global area of the missing data. For a single 73° or 82° satellite, this would lead to a systematic bias in the final global average of 0.4 $\mathrm{Wm}^{-2}$ or 0.1 $\mathrm{Wm}^{-2}$, respectively.

In order to remain within the nominal target uncertainty of 1.0 $\mathrm{Wm}^{-2}$, we could then in principle accept systematic errors of 20 $\mathrm{Wm}^{-2}$ or 80 $\mathrm{Wm}^{-2}$ for these regions. The 82° orbit thus has a clear advantage in this particular case by virtue of its higher maximum latitude. However, this threshold is only a theoretical upper bound, based purely on the sampling error and the area of the affected polar regions. In practice, additional error sources would need to be accounted for, and the maximum tolerable systematic error would be much lower.

Overall, single satellites could be a viable option, provided that missing data can be properly accounted for. However, a more straightforward solution is to include one or more additional satellites, as per the investigations in previous sections.

### 3.5  Trends

As a slightly separate topic, we will now briefly address the potential of trend determination using the studied satellites, both individually and in constellations. For reference, the 2001-2020 trend in the CERES input data used here, determined by simple linear regression, is +0.33 $\mathrm{Wm}^{-2}\mathrm{decade}^{-1}$ (net radiation) or -0.36 $\mathrm{Wm}^{-2}\mathrm{decade}^{-1}$ (total outgoing radiation). The RMS trend errors for the various satellites and constellations, over intervals of different lengths, are shown in Tables 4 and 5. Results for the case of systematic positive or negative bias in the camera-determined OSR fraction (Sect. 2.5) are not included for brevity, but change little compared with the unbiased correction, much like the case with Table 2.

As with the annual-mean results in previous sections, the error decreases as the number of satellites increases, and here extended time intervals also lead to reduced errors. Already after five years, the RMS error for all constellations is smaller

than the actual trend value, which is very promising for detection. There are a few other points that stand out among the results: First, it is worth noting that even single satellites perform reasonably well. In particular, the 90° satellites are now comparable with the 73° and 82° satellites, unlike the annual-mean results where single 90° satellites performed noticeably worse. Second, the 2x 90° constellation now performs noticeably better than the 73°+90° and 82°+90° constellations, with RMS errors reduced by at least half in comparison with those constellations. Third, the shortwave correction that resulted in clear improvements for the annual means still results in an improvement in almost all cases, but the effect is less pronounced in the trend results. Fourth, constellations of sun-synchronous satellites perform significantly better in the trend than in the annual means. In fact, the 2x 98° constellation now performs better than the 73°+90° and 82°+90° constellations. The trend errors from sun-synchronous constellations are comparable to, but still slightly greater than, those from polar constellations with the same number of satellites.

Overall, these results show that trend measurements are possible with all of the investigated constellations, and that their sampling errors allow for detectable trends within five years.

| Satellite inclination | 73°+90° | 82°+90° | 2x 90° | 3x 90° | 4x 90° | 73° | 82° | 90° |
|---|---|---|---|---|---|---|---|---|
| Ensemble size | 144 | 144 | 6 | 4 | 3 | 12 | 12 | 12 |
| 5-year interval | | | | | | | | |
| RMS error [no SW corr] | 0.29 | 0.21 | 0.08 | 0.04 | 0.03 | 0.29* | 0.21* | 0.22 |
| RMS error [SW corr] | 0.23 | 0.21 | 0.09 | 0.04 | 0.04 | 0.24* | 0.21* | 0.18 |
| 10-year interval | | | | | | | | |
| RMS error [no SW corr] | 0.16 | 0.13 | 0.03 | 0.02 | 0.01 | 0.18* | 0.13* | 0.08 |
| RMS error [SW corr] | 0.09 | 0.10 | 0.03 | 0.02 | 0.02 | 0.10* | 0.10* | 0.08 |
| 20-year interval | | | | | | | | |
| RMS error [no SW corr] | 0.11 | 0.06 | 0.01 | 0.004 | 0.003 | 0.14* | 0.06* | 0.05 |
| RMS error [SW corr] | 0.03 | 0.02 | 0.01 | 0.003 | 0.004 | 0.03* | 0.02* | 0.02 |

**Table 4.** Root mean square (RMS) error in the trend of total outgoing TOA radiation, with and without shortwave correction (SW corr), relative to the reference truth. All values are given in units $\mathrm{Wm}^{-2}\mathrm{decade}^{-1}$. These errors are computed over all ensemble members, and all non-overlapping time periods for each period length. The values are based on measurements with different constellations: one precessing satellite (73° or 82°) in combination with one polar satellite (90°) to fill in data for the poles, or two, three or four polar satellites. The results for individual satellites (73°, 82°, 90°) are included for comparison. *These results only cover the latitude range of the satellites, and thus do not include the full polar regions. The effect of missing polar data is discussed in Sect. 3.4.

| Satellite inclination | 2x 98° | 3x 98° | 4x 98° | 6x 98° |
|---|---|---|---|---|
| Ensemble size | 6 | 4 | 3 | 2 |
| 5-year interval | | | | |
| RMS error [no shortwave correction] | 0.12* | 0.11* | 0.08* | 0.05* |
| RMS error [with shortwave correction] | 0.13* | 0.10* | 0.07* | 0.04* |
| 10-year interval | | | | |
| RMS error [no shortwave correction] | 0.10* | 0.06* | 0.04* | 0.04* |
| RMS error [with shortwave correction] | 0.06* | 0.04* | 0.02* | 0.02* |
| 20-year interval | | | | |
| RMS error [no shortwave correction] | 0.03* | 0.01* | 0.02* | 0.01* |
| RMS error [with shortwave correction] | 0.05* | 0.01* | 0.02* | 0.01* |

**Table 5.** As Table 4 for constellations of sun-synchronous satellites (98°). All values are given in units $\mathrm{Wm}^{-2}\mathrm{decade}^{-1}$.

## 3.6 Future studies

The analysis above allows some conclusions to be drawn, but there are issues that remain for further investigation in future studies. As mentioned in the introduction, an investigation of angular dependence and anisotropic radiation is already planned, which should provide valuable insights into the shortwave radiation in particular. It would be reasonable to analyse twilight transmission and the associated error in more detail at the same time. To first order, the geometry of the Sun, the Earth's surface and the satellite field of view would determine the transmission, but this would in principle also be affected by the atmospheric

layer. On the satellite side of the analysis, it would be very relevant to properly investigate the effects of the instrument response on the final result, and determine quantitative requirements for the instrument performance. In terms of the data used as input, it may be worth analysing the simulated satellite performance in relation to the features of the original input, e.g. which features are detectable and carry through to the final result. Ideally, this could be done as a study of point-source radiation in different locations, in line with Green's function responses. An obvious candidate would be a set of input data that correspond to different

past or future scenarios for greenhouse gas emissions. Another potential topic would be the impact of single dramatic events, such as volcanic eruptions.

## 4 Conclusions

The EEI is a critical quantity for monitoring the climate system. Hence, an adequate measurement system that includes both interior energy and satellite components is needed in order to observe the EEI over time. Satellite-borne instruments are one valuable potential source of such measurements, but the current best satellite-only global-mean EEI estimates do not reach an absolute measurement uncertainty below $1 \ \mathrm{Wm}^{-2}$. As a result, satellite radiation measurements cannot currently independently verify EEI estimates from interior methods such as ocean temperature measurements, which monitor the changes in accumulated energy within the Earth system over time. There are therefore ongoing efforts to improve the situation, both within existing satellite missions and as part of new initiatives, and using both established and novel methods.

In this work we have focused on one kind of satellite instrument, namely wide-field-of-view radiometers, and investigated how idealised EEI estimates change due to orbital sampling effects related to the diurnal and annual cycles. We simulated satellite orbits with different inclinations and used hourly TOA radiation fields with assumed Lambertian emission to generate synthetic wide-field-of-view measurements. These measurements were processed using a coarse-grid bin method, without using any information from the diurnal or annual cycles, to produce global-mean EEI estimates. Although the introduction of a priori knowledge of these cycles could reduce the estimated error, it would inherently introduce a systematic uncertainty that would need to be accounted for. Even if such an uncertainty may be small, it is by its very nature challenging to quantify. Our ambition was therefore to investigate orbital sampling options in order to minimise the estimated error without relying on a priori information.

We show that different orbital inclinations lead to different characteristics in terms of the sampling issues, with key findings that can be summarised in five points: First, no single satellite orbit provides the spatial and temporal sampling necessary to reliably estimate the global-mean TOA net imbalance. Second, two combined 90° satellites can estimate the EEI to within an RMS error of $0.10 \ \mathrm{Wm}^{-2}$. This is improved to $0.04 \ \mathrm{Wm}^{-2}$ by three 90° satellites, and with four 90° satellites, this is possible without a shortwave correction. Third, a combination of one 90° and one either 73° or 82° satellite leads to an RMS error of $0.08 \ \mathrm{Wm}^{-2}$ or $0.10 \ \mathrm{Wm}^{-2}$, respectively. Fourth, at least two satellites are necessary to achieve an uncertainty reliably lower than the current satellite-only best estimate of the EEI. If sun-synchronous orbits are used, at least six satellites are required. The latter result could be significantly improved with a diurnal cycle model, but doing so would in turn introduce the previously mentioned systematic bias. Fifth, all investigated constellations allow for detectable trends within five years.

These results can help inform current and future efforts to directly measure the EEI with satellite-borne instruments. At the same time, it is important to be aware that there are other error sources that have not been analysed here, such as the effect of non-Lambertian radiation, radiative effects from the upper atmosphere and errors associated with the instrument measuring the radiation. Future studies to address these aspects are necessary in order to bring the combined uncertainty in direct satellite measurements of the EEI below the $1 \ \mathrm{Wm}^{-2}$ threshold.

*Code and data availability.* The software is available from the Bolin Centre Database (Hocking, 2024). The input data are available from (NASA/LARC/SD/ASDC, 2017). The satellite orbits were computed with the simplified general perturbation model SGP4 (Vallado et al., 2006; Vallado and Crawford, 2008), as implemented in the Python library python-sgp4 (Rhodes, 2023).

*Author contributions.* All authors formulated the overarching goals of the study. TH developed the software, performed the analysis and prepared the manuscript, with contributions from LM and TM.

*Competing interests.* The authors declare that they have no competing interests.

*Acknowledgements.* The authors would like to thank Donal Murtagh, Jake Gristey, Norman Loeb, Seiji Kato and Steven Dewitte for useful comments and discussions. The authors would also like to thank Peter Pilewskie and one anonymous reviewer for their comments and suggestions, which helped improve the final version of this paper. This project was funded by the European Research Council (ERC) (grant agreement no. 770765), the European Union's Horizon 2020 research and innovation program (grant agreement nos. 820829 and 101003470), and the Swedish Research Council (VR) (grant agreement no. 2022-03262). The computations were enabled by resources provided by the National Academic Infrastructure for Supercomputing in Sweden (NAISS) at the National Supercomputer Centre (NSC) partially funded by the Swedish Research Council through grant agreement no. 2022-06725. Linda Megner was supported by the Swedish National Space Agency.

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
