# Peer review of "Sampling the diurnal and annual cycles of the Earth's energy imbalance with constellations of satellite-borne radiometers"

_EGUsphere, 2024_

## Referee Comment (RC1)

Review of "Sampling the diurnal and annual cycles of the Earth's energy imbalance with constellations of satellite-borne radiometers" by Hocking et al.

This paper addresses a concept to derive Earth's radiative energy imbalance from space-based observations. As the title suggests, the focus is on diurnal and annual sampling, the former being an important source of uncertainty in current Earth radiation budget observations. The authors' propose to reduce diurnal sampling errors by choosing appropriate orbits and implementing a constellation of satellites, each member of which will fly a suit of instruments that will alternate between solar and terrestrial viewing.

On L. 70 the authors' are careful to point out that "… we focus only on sampling errors using the Earth-facing radiometer, with the intention of prioritising the accuracy of the long-term global mean over spatial and temporal resolution." Idealized instruments with perfect response are assumed along with idealized Lambertian scattering and emission – only the sampling biases due to orbital characteristics of the hypothetical constellation are addressed in this paper.

The heart of the paper, the results presented in section 3 that address the attributes of polar and precessing orbits for minimizing sampling errors in acquiring EEI are sound, if not necessarily surprising. The combination of particular precessing and polar orbits have intrinsic strengths that mitigate the weakness of the others. Perhaps I missed it: presumably, the authors were motivated to derive a concept with a just a small (three) number of orbits and that is fine. Perhaps they want to state their motivation for the number of orbits. Cost? With miniaturization of sensors and spacecraft, the opportunities to deploy larger (in number) constellations will increase in time. The paper demonstrates advantages in numbers, especially in beating down sampling error. Potentially flying even larger constellations could be discussed in addition to the specific three-orbit concept proposed for their mission.

Based on the emphasis on reducing diurnal sampling errors, I recommend the paper for publication. However, I have a list of issues below that should be addressed. One important issue is the authors' target uncertainty that is the same magnitude of current EEI estimates, 1.0 Wm$^{-2}$. More than once they list that as the desired uncertainty. Granted, most of the sampling errors listed in Table 2 are on the order of 0.1 Wm$^{-2}$ but this is only one contribution to total uncertainty, many more of which will be left to future publications. To achieve the goal of resolving a 1.0 Wm$^{-2}$ difference between the incoming and outgoing radiative energy is a daunting task that requires exquisite accuracy across all elements of the measurements and analysis equations. Even in this paper the authors seemingly overlooked some error sources, for example, those due to deriving a global mean outgoing irradiance (section 2.5). I would like to see the authors address these issues below prior to publication. I selected "major revisions" but these are somewhere in between major and minor: important but not difficult to address.

Here are specific comments:

1. L. 47-48: "Current estimates of the EEI are based on both direct measurements by satellite radiometers and inventories from ocean heat content measurements."

   I suggest removing the word "direct" since the authors later explain some of the modeling required to covert directly measured radiance from satellite radiometers to global-averaged irradiance. Those are far-removed from "direct".

2. L. 65: "…for an absolute accuracy of the annual EEI within 1.0 Wm$^{-2}$."

   If the imbalance is estimated to be 1.0 Wm$^{-2}$, how is that uncertainty (and it is uncertainty, not accuracy) sufficient to resolve the EEI? To resolve that level of imbalance, far lower uncertainty is required.

3. L. 77-79: "Previous efforts have typically relied on a diurnal model to synthesise full sampling of the diurnal cycle, which requires that the model does not introduce additional errors."

   Please provide a reference.

4. L. 111: "…irradiance I…" should be "…radiance I…"

5. L. 112: "Taking into account the $1/d^2$ decrease of the intensity with distance d …"

   Change "intensity" to "irradiance" since intensity is undefined in the paper and it is no longer a standard term in radiative transfer.

6. Eq. 1 is an interesting form of the equation relating radiance to irradiance; you may want to point out that this form is necessary, with integration over area rather than solid angle, since you are using CERES results for $M$ (irradiance).

7. L. 117: "For a non-perfect instrument response, the ideal cos(η) factor would be adjusted accordingly."

   Please explain; "would be adjusted accordingly" for what purpose? To provide a correction to measured $F$? That is impossible because it would require a priori knowledge of $M$ (or equivalently, radiance).

8. L. 139: "… perfect cosine response to the flux"

   There is no cosine response to "flux" (and the correct term is flux density, or irradiance); irradiance is the integral over solid angle of cosine-weighted radiance.

9. Table 1: I do not think this table adds much to the paper since most the table entries are undefined. Is it possible to add at least a short description of the terms? If not, please consider removing.

10. Fig. 7 caption: "The shortwave component includes the correction described in Sect. 2.5."

    I must have missed it but I do not know what the referenced correction is; Fig. 7 is already in section 2.5

    Okay, I see that the correction comes two pages later, in equation (2). This is awkward and likely to be confusing to most readers. Please rearrange to introduce the correction prior to figure 7.

11. L. 182: "For this study, a 5° by 5° grid provided sufficiently good results …"

    Please quantify "sufficiently good".

12. L. 183: A fortnight is not a common term for all readers; please use days, weeks, etc.

13. L. 196-194: "Note that this correction requires a method to separate the full-spectrum radiometer measurements of outgoing radiation into longwave and shortwave components, such as the proposed ECO cameras"

    Since this correction relies on separation scattered sunlight from emitted terrestrial radiation, some estimate of the uncertainty using cameras to do this separation needs to be addressed.

14. L. 201-217: Similar to the previous comment, is not clear if the uncertainty due the processing required to obtain global means has been considered. The error due to the shortwave correction in particular should be estimated.

15. L. 304: "In order to remain within the nominal target uncertainty of 1.0 Wm−2 …"

    As in comment 1, it is confusing that the target uncertainty is 100% of the estimated imbalance.

16. L. 315-316: "As a result, satellite radiation measurements cannot currently independently verify EEI estimates from interior methods."

    Do  "interior measurements" refer to the Argo float network of ocean temperature measurements (more commonly called *in situ*)?

---

## Referee Comment (RC2)

General comments:
This manuscript is well written, clear, and a useful contribution to the field of Earth Radiation Budget research and observations. It clearly is in support of the science objectives of the ESA ECO mission that aims at high accuracy EEI measurements over annual to multi-annual timescales employing WFOV radiometry. This study isolates the problem of space-time sampling from other sources of uncertainty pertaining to instrumentation and anisotropy of the radiance field. The insights gained, support previous findings that sampling of diurnal and intra-annual variability is critical for ERB research. That said, the manuscript lacks a bit of background on the issues at hand and previous studies. I believe it would be beneficial to highlight (some of) the history of orbital constellation studies and put the new findings into perspective. Overall, I recommend this paper for publication. I hope the authors find my comments and suggestions below helpful for improving their manuscript

Specific Comments:

1) Line 65: Does the suggested 1 $Wm^{-2}$ accuracy requirement meet the science needs? Who established this requirement and how/why?
2) Line 69: It is unclear to me how the camera "allows" to distinguish spatial resolution. I assume the authors are referring to sub-footprint variability that the camera resolves to some extent? What is the spatial/spectral resolution of the camera and how will it be used?
3) Line 77 ff: That the diurnal cycle represents an issue in sampling regional and global ERB correctly is a known fact. Likewise, inclined (precessing) orbits have been suggested by many studies to improve on this issue. There are likely many more studies on this. I was able to find these:

- Kirk-Davidoff, D. B., R. M. Goody, and J. G. Anderson, 2005: Analysis of Sampling Errors for Climate Monitoring Satellites. *J. Climate*, **18**, 810–822, https://doi.org/10.1175/JCLI-3301.1.
- T. H. V. Haar, T. H., and E. A. Smith, E.A. (1979). Theoretical comparison between radiometric and radiation pressure measurements for determination of the Earth's radiation budget, Atmos. Sci. Paper 317, Jul. 1979.
- Campbel and Vonder Haar (1978); cited in above
- Taylor, P. C., and N. G. Loeb, 2013: Impact of Sun-Synchronous Diurnal Sampling on Tropical TOA Flux Interannual Variability and Trends. *J. Climate*, **26**, 2184–2191, https://doi.org/10.1175/JCLI-D-12-00416.1.
- Salby, M. L., 1988: Asynoptic Sampling Considerations for Wide-Field-of-View Measurements of Outgoing Radiation. Part I: Spatial and Temporal Resolution. *J. Atmos. Sci.*, **45**, 1176–1183, https://doi.org/10.1175
- https://ntrs.nasa.gov/api/citations/20140006546/downloads/20140006546.pdf and other works by Harrison.

4) Line 110 ff: Is there a reference for the "kernel" and equation 1? I'm sure there are several. For example, papers that intercompared ERBE WFOV and scanner data back in the 1980s/90s.

5) Line 116: Is it shape factor or anisotropy factor? What is the difference? And where can the reader look up background information (reference)?

6) Line 117: What are typical cosine response errors and how would they affect the measurement? This simulation environment would be perfect for quantifying the requirements for these errors and instrument response. These errors cannot be corrected for once the measurement is taken. I'm wondering, however, if with this model the error scan be predicted using CERES or camera data to correct for it during data processing. Of course, this won't be perfect either, but might become necessary.

7) Figure 4: What are the corresponding global mean OLR values? Is the right figure the same as left but multiplied with the kernel?

8) Line134: Since EEI is to be measured at high accuracy, what is the magnitude of atmospheric twilight transmission and the error induced? I'm wondering if this work by Loeb et al., 2002 might provide insight: https://journals.ametsoc.org/view/journals/clim/15/22/1520-0442_2002_015_3301_dtotaf_2.0.co_2.xml

9) Line 210: Please clarify what is ISR vs $ISR_{CERES}$. Is $ISR_{CERES}$ the truth if perfectly sampled? And is ISR the undersampled measurement?

10) Lines 234-235: I believe the opposite may be true. The more satellites, the less susceptible the mission and data record is to loss of instruments. As mentioned earlier, the diurnal filling can be achieved in other ways. 6 satellites do not seem that impractical (e.g. compared to the Irridium66 example), and fixed local times (SSO) may have many advantages, e.g., well known return time and a better handle on intercalibration targets. It really depends on the needs of the mission and trade space. I would not completely disregard a SSO constellation. There are reasons why most Earth science missions fly in SSO and there may be more opportunity for reaching such orbits, e.g., on ride shares if needed.

11) Line 237: Even though the errors seem small when using the 2 or 3 sat constellation, the sampling of Earth is still far from complete. For example, what if a major event such as a volcanic eruption occurs? The CERES record does not cover any such event. This would be a good experiment to conduct. In general, this paper should end on "next steps" that will be taken to improve the model, and additional analysis that will be conducted to answer any remaining questions.

12) Line 255: What about even lower inclinations, e.g., 68deg? This would increase the sampling of diurnal cycle even more. Do you know at which inclination the benefit of diurnal sampling goes to near zero? Such a sensitivity study would be very useful. Previous studies suggested inclinations near 60 and 50 deg, probably to enhance the sampling specifically at low latitudes where it is most significant.

Technical Comments:

- Abstract line 5: "There has recently been a renewed interest in applying wide-field-of-view radiometers onboard satellites to measure the outgoing radiation, and hence deduce the global annual mean energy imbalance." – It is unclear to me how one can deduce EEI from Earth out going radiation alone. I recommend this sentence to be rewritten.
- Line 16: A number of papers could be cited here after the first sentence, e.g., Loeb et al., 2021; Raghuraman et al. 2021; Kramer et al., 2021…
- Line 28: I believe there is consensus that the "solar constant" is not a constant at all. "Total Solar Irradiance (at 1 AU)" would be more fitting.
- Line 33: Stephens et al. (2015) provide a history of albedo values and studies.
- https://agupubs.onlinelibrary.wiley.com/doi/10.1002/2014RG000449
- Line 40: "Spread" might not be the proper wording. Do you mean "combination of"?
- Line 46: Hakuba et al., 2021 is also a good example of satellite-based ocean heat uptake and change in EEI deduced from it.
  https://agupubs.onlinelibrary.wiley.com/doi/full/10.1029/2021GL093624
- Line 53: I recommend these references for the Libera mission:
  - Harber, D., K. Catani, J. Gieseler, R. Haun, N. Kruczek, J. Sprunk, N. Tomlin, C. Yung, J. Lehman, M. Stephens, T. Kampe, S. Collins, J. Peterson, H. Latvakoski, C. Monte, M. Hakuba, and P. Pilewskie (2013). The Libera Mission: Bringing Next-Generation Technology to an Established Climate Data Record. 15th International Conference on New Developments and Applications in Optical Radiometry (NEWRAD 2023), 11–15 Sep. 2023, NPL, Teddington, UK.
  - Hakuba et al. (2024): Maria Z. Hakuba, Bruce Kindel, Jake Gristey, Alejandro Bodas-Salcedo, Graeme Stephens, Peter Pilewskie; Simulated variability in visible and near-IR irradiances in preparation for the upcoming Libera mission. *AIP Conf. Proc.* 18 January 2024; 2988 (1): 050006. https://doi.org/10.1063/5.0183869.

---

## Author Comment (AC1)

**Response to reviewers**

Thomas Hocking, Thorsten Mauritsen, Linda Megner

**Introduction**

We are grateful to Peter Pilewskie and an anonymous referee for their comments on the manuscript "Sampling the diurnal and annual cycles of the Earth's energy imbalance with constellations of satellite-borne radiometers" (egusphere-2024-356).

Overarching comments are given immediately below. Responses to specific points from the two reviews are provided on the following pages, with the original comments in black and our responses in blue. A full record of the exact changes in the revised document are available as a manuscript with tracked changes, uploaded separately. Due to technical issues, the changes in tables and in the "Code and data availability" section are unfortunately very hard to read or not visible, in which case we refer interested readers to the revised manuscript without tracked changes.

**Errors and uncertainties**

Both reviewers commented on errors and uncertainties, and requested further detail and clarification. We realise that the original manuscript was somewhat confusing on this topic, and we have made changes in various parts of the document. Rather than a formal error propagation through the whole analysis sequence, we present statistical uncertainties based on the results of ensembles of simulated constellations, and have performed some sensitivity tests to address issues raised by the reviewers. To facilitate interpretation, we also adjusted the values in Table 2 to use RMS errors. Because of this adjusted calculation, the resulting numerical values have changed slightly, but the overall conclusions have not.

**Larger constellations**

Additionally, both reviewers commented on larger constellations. Our main focus remains on small constellations of non-sun-synchronous satellites, but we have added analysis of a 4-satellite polar constellation as well as sun-synchronous constellations with 2, 3, 4 and 6 satellites.

**Trends**

The original manuscript only considered annual values. In the revised manuscript, we have also included a brief analysis of interannual trends in Section 3.5.

**Comments by Peter Pilewskie**

Review of "Sampling the diurnal and annual cycles of the Earth's energy imbalance with constellations of satellite-borne radiometers" by Hocking et al.

This paper addresses a concept to derive Earth's radiative energy imbalance from space-based observations. As the title suggests, the focus is on diurnal and annual sampling, the former being an important source of uncertainty in current Earth radiation budget observations. The authors' propose to reduce diurnal sampling errors by choosing appropriate orbits and implementing a constellation of satellites, each member of which will fly a suit of instruments that will alternate between solar and terrestrial viewing.

On L. 70 the authors' are careful to point out that ". . . we focus only on sampling errors using the Earth-facing radiometer, with the intention of prioritising the accuracy of the long-term global mean over spatial and temporal resolution." Idealized instruments with perfect response are assumed along with idealized Lambertian scattering and emission – only the sampling biases due to orbital characteristics of the hypothetical constellation are addressed in this paper.

The heart of the paper, the results presented in section 3 that address the attributes of polar and precessing orbits for minimizing sampling errors in acquiring EEI are sound, if not necessarily surprising. The combination of particular precessing and polar orbits have intrinsic strengths that mitigate the weakness of the others. Perhaps I missed it: presumably, the authors were motivated to derive a concept with a just a small (three) number of orbits and that is fine. Perhaps they want to state their motivation for the number of orbits. Cost? With miniaturization of sensors and spacecraft, the opportunities to deploy larger (in number) constellations will increase in time. The paper demonstrates advantages in numbers, especially in beating down sampling error. Potentially flying even larger constellations could be discussed in addition to the specific three-orbit concept proposed for their mission. Even though opportunities for larger constellations can be expected to improve in time, for now the prospect of more than three satellites was considered to be limited and hence of limited relevance for the current study. Furthermore, we feel that the results for single satellites or two-satellite constellations better illustrate the relevant sampling issues. Nevertheless, we have added a brief discussion of constellations with four polar satellites or up to six sun-synchronous satellites.

Based on the emphasis on reducing diurnal sampling errors, I recommend the paper for publication. However, I have a list of issues below that should be addressed. One important issue is the authors' target uncertainty that is the same magnitude of current EEI estimates, 1.0 Wm-2. More than once they list that as the desired uncertainty. Granted, most of the sampling errors listed in Table 2 are on the order of 0.1 Wm-2 but this is only one contribution to total uncertainty, many more of which will be left to future publications. To achieve the goal of resolving a 1.0 Wm-2 difference between the incoming and outgoing radiative energy is a daunting task that requires exquisite accuracy across all elements of the measurements and analysis equations. Even in this paper the authors seemingly overlooked some error sources, for example, those due to deriving a global mean outgoing irradiance (section 2.5). I would like to see the authors address these issues below prior to publication. I selected "major revisions" but these are somewhere in between major and minor: important but not difficult to address.

Here are specific comments:

1. L. 47-48: "Current estimates of the EEI are based on both direct measurements by satellite radiometers and inventories from ocean heat content measurements."
   I suggest removing the word "direct" since the authors later explain some of the modeling required to covert directly measured radiance from satellite radiometers to global-averaged irradiance. Those are far-removed from "direct".
   Removed "direct"

2. L. 65: ". . . for an absolute accuracy of the annual EEI within 1.0 Wm-2."
   If the imbalance is estimated to be 1.0 Wm-2, how is that uncertainty (and it is uncertainty, not accuracy) sufficient to resolve the EEI? To resolve that level of imbalance, far lower uncertainty is required.
   We acknowledge that the exact uncertainty required to definitively detect or resolve a given imbalance depends on the level of confidence and the number of measurements that are available. As these are not fixed, we consider the natural choice of target uncertainty to be the same value as the target imbalance, despite the resulting ambiguity.
   Changed "accuracy" to "uncertainty"

3. L. 77-79: "Previous efforts have typically relied on a diurnal model to synthesise full sampling of the diurnal cycle, which requires that the model does not introduce additional errors."
Please provide a reference.
Done

4. L. 111: "...irradiance I..." should be "...radiance I..."
Changed "irradiance" to "radiance"

5. L. 112: "Taking into account the $1/d^2$ decrease of the intensity with distance d ..."
Change "intensity" to "irradiance" since intensity is undefined in the paper and it is no longer a standard term in radiative transfer.
Changed "intensity" to "irradiance at the measurement location"

6. Eq. 1 is an interesting form of the equation relating radiance to irradiance; you may want to point out that this form is necessary, with integration over area rather than solid angle, since you are using CERES results for M (irradiance).
This has been clarified.

7. L. 117: "For a non-perfect instrument response, the ideal $\cos(\eta)$ factor would be adjusted accordingly."
Please explain; "would be adjusted accordingly" for what purpose? To provide a correction to measured F? That is impossible because it would require a priori knowledge of M (or equivalently, radiance).
The quoted sentence was intended as a general statement about the physical significance of this factor and the difference compared with a real instrument, rather than as e.g. a potential correction method. This has been rephrased and elaborated upon.

8. L. 139: "... perfect cosine response to the flux"
There is no cosine response to "flux" (and the correct term is flux density, or irradiance); irradiance is the integral over solid angle of cosine-weighted radiance.
This has been rewritten

9. Table 1: I do not think this table adds much to the paper since most the table entries are undefined. Is it possible to add at least a short description of the terms? If not, please consider removing.
Short descriptions have been added.

10. Fig. 7 caption: "The shortwave component includes the correction described in Sect. 2.5."
I must have missed it but I do not know what the referenced correction is; Fig. 7 is already in section 2.5
Okay, I see that the correction comes two pages later, in equation (2). This is awkward and likely to be confusing to most readers. Please rearrange to introduce the correction prior to figure 7.
This has been rearranged somewhat. We agree that it is awkward, but feel that the sequence of processing steps, including the correction, is nevertheless more appropriate after the figure. To mitigate the issue, Figure 7 has been moved further down in the document, and the caption reference now explicitly points to equation 2 "near the end of Sect. 2.5".

11. L. 182: "For this study, a 5° by 5° grid provided sufficiently good results ..."
Please quantify "sufficiently good".
This has been rephrased and clarified

12. L. 183: A fortnight is not a common term for all readers; please use days, weeks, etc.
Changed "fortnight" to "14 days".

13. L. 196-194: "Note that this correction requires a method to separate the full-spectrum radiometer measurements of outgoing radiation into longwave and shortwave components, such as the proposed ECO cameras"
Since this correction relies on separation scattered sunlight from emitted terrestrial radiation, some estimate of the uncertainty using cameras to do this separation needs to be addressed.
We do not consider the camera performance in detail, but consider it reasonable to assume that the shortwave/total fraction can be determined to within $\pm10\%$ or better. The resulting uncertainty is determined as part of a sensitivity test of the shortwave correction (see next point).

14. L. 201-217: Similar to the previous comment, is not clear if the uncertainty due the processing required to obtain global means has been considered. The error due to the shortwave correction in particular should be estimated.

   We have performed a sensitivity test with $\pm 10\%$ biases in the measured shortwave fraction, and included these results in Table 2. We consider the uncertainty of the whole chain of processing and analysis as a single unit, and quantify it in terms of statistical uncertainties based on the ensemble results.

15. L. 304: "In order to remain within the nominal target uncertainty of 1.0 Wm-2 ..."

   As in comment 1, it is confusing that the target uncertainty is 100% of the estimated imbalance.

   Please see response to earlier comment

16. L. 315-316: "As a result, satellite radiation measurements cannot currently independently verify EEI estimates from interior methods."

   Do "interior measurements" refer to the Argo float network of ocean temperature measurements (more commonly called in situ)?

   This has been clarified

**Comments by Anonymous Referee #2**

General comments:

This manuscript is well written, clear, and a useful contribution to the field of Earth Radiation Budget research and observations. It clearly is in support of the science objectives of the ESA ECO mission that aims at high accuracy EEI measurements over annual to multi-annual timescales employing WFOV radiometry. This study isolates the problem of space-time sampling from other sources of uncertainty pertaining to instrumentation and anisotropy of the radiance field. The insights gained, support previous findings that sampling of diurnal and intra-annual variability is critical for ERB research. That said, the manuscript lacks a bit of background on the issues at hand and previous studies. I believe it would be beneficial to highlight (some of) the history of orbital constellation studies and put the new findings into perspective. Overall, I recommend this paper for publication. I hope the authors find my comments and suggestions below helpful for improving their manuscript

We have included some additional background, with reference to previous studies for perspective.

Specific Comments:

1. Line 65: Does the suggested 1 Wm-2 accuracy requirement meet the science needs? Who established this requirement and how/why?
   This value was based on the ECO proposal, which lists a target uncertainty "better than 1.0 W/m2". There is some ambiguity regarding what a specific uncertainty can actually resolve, depending on the intended level of confidence and the expected number of measurements, but qualitatively, this would address the statistical issue of determining the sign of the true imbalance. With a lower annual uncertainty, which our results show is possible, and measurements over multiple years, it would naturally be possible to achieve even better results.

2. Line 69: It is unclear to me how the camera "allows" to distinguish spatial resolution. I assume the authors are referring to sub-footprint variability that the camera resolves to some extent? What is the spatial/spectral resolution of the camera and how will it be used?
   The camera should indeed be able to resolve some of the sub-footprint variability, thanks to the pixels of the camera. However, this is not used in the current study, and so we do not address the spatial resolution further.
   The spectral resolution is fundamentally based on separate longwave and shortwave measurements, with increased resolution through a combination of multiple sensors and narrow-band filters. For the current study, the outgoing radiation only needs to be separated into shortwave and longwave components. As such, a detailed description and analysis of the spectral performance of the cameras is outside the scope of our investigation. Nevertheless, we have performed a more general sensitivity test based on the measured shortwave fraction, and included these results in Tables 2 and 3.

3. Line 77 ff: That the diurnal cycle represents an issue in sampling regional and global ERB correctly is a known fact. Likewise, inclined (precessing) orbits have been suggested by many studies to improve on this issue. There are likely many more studies on this. I was able to find these:
   This has been expanded to better cover previous studies, with references.

   - Kirk-Davidoff, D. B., R. M. Goody, and J. G. Anderson, 2005: Analysis of Sampling Errors for Climate Monitoring Satellites. J. Climate, 18, 810–822,
     `https://doi.org/10.1175/JCLI-3301.1`.
   - T. H. V. Haar, T. H., and E. A. Smith, E.A. (1979). Theoretical comparison between radiometric and radiation pressure measurements for determination of the Earth's radiation budget, Atmos. Sci. Paper 317, Jul. 1979.
   - Campbel and Vonder Haar (1978); cited in above
   - Taylor, P. C., and N. G. Loeb, 2013: Impact of Sun-Synchronous Diurnal Sampling on Tropical TOA Flux Interannual Variability and Trends. J. Climate, 26, 2184–2191,
     `https://doi.org/10.1175/JCLI-D-12-00416.1`.
   - Salby, M. L., 1988: Asynoptic Sampling Considerations for Wide-Field-of-View Measurements of Outgoing Radiation. Part I: Spatial and Temporal Resolution. J. Atmos. Sci., 45, 1176–1183,
     `https://doi.org/10.1175`
   - `https://ntrs.nasa.gov/api/citations/20140006546/downloads/20140006546.pdf` and other works by Harrison.

4. Line 110 ff: Is there a reference for the "kernel" and equation 1? I'm sure there are several. For example, papers that intercompared ERBE WFOV and scanner data back in the 1980s/90s.
   This passage has been adjusted, and references have been added.

5. Line 116: Is it shape factor or anisotropy factor? What is the difference? And where can the reader look up background information (reference)?
   Corrected "shape factor" to "anisotropy factor", and expanded on the difference between the two.

6. Line 117: What are typical cosine response errors and how would they affect the measurement? This simulation environment would be perfect for quantifying the requirements for these errors and instrument response. These errors cannot be corrected for once the measurement is taken. I'm wondering, however, if with this model the error scan be predicted using CERES or camera data to correct for it during data processing. Of course, this won't be perfect either, but might become necessary.
   We agree that this is a very interesting topic, and the simulation environment is in principle very promising for such an investigation. Unfortunately, we feel that this is outside the scope of the current manuscript, but we are considering this for a future study. Nevertheless, we have added some qualitative statements on the effect of these errors on the measurement.

7. Figure 4: What are the corresponding global mean OLR values? Is the right figure the same as left but multiplied with the kernel?
   The right figure is indeed computed from the left figure using Equation 1. This has been clarified and the corresponding global mean OLR values have been added to the figure caption.

8. Line134: Since EEI is to be measured at high accuracy, what is the magnitude of atmospheric twilight transmission and the error induced? I'm wondering if this work by Loeb et al., 2002 might provide insight:
   `https://journals.ametsoc.org/view/journals/clim/15/22/1520-0442_2002_015_3301_dtotaf_2.0.co_2.xml`
   This is indeed an issue that should be analysed. We currently treat the Earth as an emitting shell that exactly fills the idealised satellite field of view, and as such the sampling is not affected by twilight transmission. Therefore, this issue is outside the scope of our current analysis, but we intend to investigate this further in a future publication. This limitation has been clarified in the revised manuscript.

9. Line 210: Please clarify what is ISR vs $ISR_{CERES}$. Is $ISR_{CERES}$ the truth if perfectly sampled? And is ISR the undersampled measurement?
   This has been clarified

10. Lines 234-235: I believe the opposite may be true. The more satellites, the less susceptible the mission and data record is to loss of instruments. As mentioned earlier, the diurnal filling can be achieved in other ways. 6 satellites do not seem that impractical (e.g. compared to the Irridium66 example), and fixed local times (SSO) may have many advantages, e.g., well known return time and a better handle on intercalibration targets. It really depends on the needs of the mission and trade space. I would not completely disregard a SSO constellation. There are reasons why most Earth science missions fly in SSO and there may be more opportunity for reaching such orbits, e.g., on ride shares if needed.
    It is true that an increased number of satellites generally increases resilience to instrument loss, and our initial argument was not well phrased. Even though we explicitly do not address instrument performance in this manuscript, it is clear that the instrument requirements to achieve the desired overall uncertainty would be strict. In this context, a six-satellite science mission with adequate instrumentation would be a tall order. In our idealised framework, it is also difficult to integrate the practical benefits connected to intercalibration targets and ride shares. Our results illustrate some of the issues with SSO constellations, compared with the non-SSO constellations that we later present. Because of our desire to limit the need for diurnal filling, we thus focused our attention on these non-SSO constellations that provided better results with fewer satellites. This section has been simplified, and a brief discussion of SSO constellations has been added in Section 3.3.

11. Line 237: Even though the errors seem small when using the 2 or 3 sat constellation, the sampling of Earth is still far from complete. For example, what if a major event such as a volcanic eruption occurs? The CERES record does not cover any such event. This would be a good experiment to conduct. In general, this paper should end on "next steps" that will be taken to improve the model, and additional analysis that will be conducted to answer any remaining questions.

In order to noticeably affect the true global annual mean that we are interested in, a volcanic eruption or other major event would need to cause a change in the radiation balance with large amplitude, duration and/or geographical extent. It is unfortunate for the analysis that the CERES record does not cover something like this. For future studies, we will consider point-source perturbations or other experiments to investigate the effect on the sampling.

We have added a new subsection at the end of section 3 to address the next steps.

12. Line 255: What about even lower inclinations, e.g., 68deg? This would increase the sampling of diurnal cycle even more. Do you know at which inclination the benefit of diurnal sampling goes to near zero? Such a sensitivity study would be very useful. Previous studies suggested inclinations near 60 and 50 deg, probably to enhance the sampling specifically at low latitudes where it is most significant.

We agree that such a sensitivity study would be useful. A highly simplified analysis indicates that the maximum benefit is found for an inclination somewhere in the 55°-85° range (Fig. 1, below), which is consistent with previous studies. Unfortunately, we have not carried out an investigation of the full range of inclinations within our full simulation framework, but we feel that the 73° and 82° inclinations nevertheless provide relevant information about the benefits of diurnal sampling in combination with a polar satellite. A comment has been added to the manuscript.

Technical Comments:

- Abstract line 5: "There has recently been a renewed interest in applying wide-field-of-view radiometers on-board satellites to measure the outgoing radiation, and hence deduce the global annual mean energy imbalance." – It is unclear to me how one can deduce EEI from Earth out going radiation alone. I recommend this sentence to be rewritten.
  This sentence has been rewritten.

- Line 16: A number of papers could be cited here after the first sentence, e.g., Loeb et al., 2021; Raghuraman et al. 2021; Kramer et al., 2021...
  Citations have been added

- Line 28: I believe there is consensus that the "solar constant" is not a constant at all. "Total Solar Irradiance (at 1 AU)" would be more fitting.
  This has been changed

- Line 33: Stephens et al. (2015) provide a history of albedo values and studies.
  `https://agupubs.onlinelibrary.wiley.com/doi/10.1002/2014RG000449`
  A reference to this paper has been added for further details.

- Line 40: "Spread" might not be the proper wording. Do you mean "combination of"?
  This has been reworded

- Line 46: Hakuba et al., 2021 is also a good example of satellite-based ocean heat uptake and change in EEI deduced from it.
  `https://agupubs.onlinelibrary.wiley.com/doi/full/10.1029/2021GL093624`
  This reference has been added

- Line 53: I recommend these references for the Libera mission:
  The Hakuba et al. reference has been included.
  The Harber et al. reference does seem very relevant, but has unfortunately not yet been published, according to the first author. As such, in line with AMT instructions, we do not include this reference. Depending on its publication status and the revision process of the current manuscript, it may be included in a later revision.

  - Harber, D., K. Catani, J. Gieseler, R. Haun, N. Kruczek, J. Sprunk, N. Tomlin, C. Yung, J. Lehman, M. Stephens, T. Kampe, S. Collins, J. Peterson, H. Latvakoski, C. Monte, M. Hakuba, and P. Pilewskie (2013). The Libera Mission: Bringing Next-Generation Technology to an Established Climate Data Record. 15th International Conference on New Developments and Applications in Optical Radiometry (NEWRAD 2023), 11–15 Sep. 2023, NPL, Teddington, UK.
  - Hakuba et al. (2024): Maria Z. Hakuba, Bruce Kindel, Jake Gristey, Alejandro Bodas-Salcedo, Graeme Stephens, Peter Pilewskie; Simulated variability in visible and near-IR irradiances in preparation for the upcoming Libera mission. AIP Conf. Proc. 18 January 2024; 2988 (1): 050006.
    https://doi.org/10.1063/5.0183869.

[Figure]

Figure 1: Profile of global-mean RMS error in outgoing radiation for filled-pole constellations with one polar and one non-polar satellite. This is based on highly simplified assumptions: each satellite can perfectly measure the true zonal mean profile for its observed local time, which changes each day. For latitude values beyond the chosen inclination, values from the corresponding profile of a polar satellite are used instead, as in the filled-pole results in the manuscript. Satellites are initialised at twelve different local times, and results are computed for the ensemble of 144 members.

This simplified analysis does not take into account the smoothing effect of the WFOV radiometer footprint, and the values are not directly comparable to those in the manuscript. However, the profile should be qualitatively indicative of the expected profile if using the complete manuscript method. Based on the footprint cumulative response function (manuscript Fig. 3), it is reasonable to assume that the stationary point of the diurnal sampling benefit would occur within ∼15° of the simplified profile minimum, i.e. ∼55°-85°. The corresponding RMS values from the revised manuscript are shown for comparison, and indicate that the amplitude of the smoothed profile would be smaller than the simplified profile.